# Interannual Variability of Summer Surface Mass Balance and Surface Melting in the Amundsen Sector, West Antarctica

Marion Donat-Magnin[1], Nicolas C. Jourdain[1], Hubert Gallée[1], Charles Amory[3], Christoph Kittel[3], Xavier Fettweis[3], Jonathan D. Wille[1], Vincent Favier[1], Amine Drira[1], Cécile Agosta[2].

[1]Université Grenoble Alpes/CNRS/IRD/G-INP, IGE, 38000, Grenoble, France
[2] Laboratoire des Sciences du Climat et de l'Environnement, IPSL/CEA-CNRS-UVSQ UMR 8212, CEA Saclay, F-91190, Gif-sur-Yvette, France
[3]F.R.S.-FNRS, Laboratory of Climatology, Department of Geography, University of Liège, B-4000 Liège, Belgium

*Correspondence to*: Marion Donat-Magnin (marion.donatmagnin@gmail.com)

**Abstract.**

*Understanding the interannual variability of Surface Mass Balance (SMB) and surface melting in Antarctica is key to quantify the signal to noise ratio in climate trends, identify opportunities for multi-year climate predictions, and to assess the ability of climate models to respond to climate variability. Here we simulate summer SMB and surface melting from 1979 to 2017 using the regional atmospheric model MAR at 10 km resolution over the drainage basins of the Amundsen Sea glaciers in West Antarctica. Our simulations reproduce the mean present-day climate in terms of near-surface temperature (mean overestimation of 0.10 °C), near-surface wind speed (mean underestimation of 0.42 m s$^{-1}$), and SMB (relative bias < 20% over Thwaites glacier). The simulated interannual variability of SMB and melting is also close to observation-based estimates.*

*For all the Amundsen glacial drainage basins, the interannual variability of summer SMB and surface melting are driven by two distinct mechanisms: high summer SMB tends to occur when the Amundsen Sea Low (ASL) is shifted southward and westward, while high summer melt rates tend to occur when ASL is shallower (i.e. anticyclonic anomaly). Both mechanisms create a northerly flow anomaly that increases moisture convergence and cloud cover over the Amundsen Sea and therefore favors snowfall and downward longwave radiation over the ice sheet. The part of interannual summer SMB variance explained by the ASL longitudinal migrations increases westward and reaches 40% for Getz. Interannual variation of the ASL relative central pressure is the largest driver of melt-rate variability, with 11 to 21% of explained variance (increasing westward). While high summer SMB and melt rates are both favored by positive phases of El Niño Southern Oscillation (ENSO), the Southern Oscillation Index (SOI) only explains 5 to 16 % of SMB or melt rates interannual variance in our simulations, with moderate statistical significance. However, the part explained by SOI in the previous austral winter is greater, suggesting that at least a part of the ENSO-SMB and ENSO-melt relationships in summer is inherited from the previous austral winter. Possible mechanisms involve sea-ice advection from the Ross Sea and intrusions of circumpolar deep water combined with melt-induced ocean overturning circulation in ice-shelf cavities. Finally, we do not find any correlation with the Southern Annular Mode (SAM) in summer.*

## 1 Introduction

From 1992 to 2017, the Antarctic continent has contributed 7.6 ± 3.9 mm to the global mean sea level (Shepherd et al., 2018) and this contribution may increase over the next century (Ritz et al., 2015; DeConto and Pollard, 2016; Edwards et al., 2019). The recent mass loss from the Antarctic ice sheet is dominated by increased ice discharge into the ocean (Shepherd et al., 2018), but both surface mass balance (SMB) and ice discharge may significantly affect the Antarctic contribution to future sea level rise (Asay-Davis et al., 2017; Favier et al., 2017; Pattyn et al., 2018). Despite recent improvements of ice-sheet models motivated by newly available satellite products over the last 10-20 years, large uncertainties remain in both the SMB and ice dynamics projections, hampering our ability to accurately predict future sea level rise (Favier et al., 2017; Shepherd and Nowicki, 2017).

Largest ice discharge changes in Antarctica are observed in the Amundsen sector with an increase of 77% over the last decades (Mouginot et al., 2014). Current changes in the dynamics of glaciers flowing into the Amundsen Sea are dominated by ocean warming rather than changes in surface conditions over the ice sheet (Thoma et al., 2008; Pritchard et al., 2012; Turner et al., 2017; Jenkins et al., 2016, 2018). Increased oceanic melting can trigger marine ice sheet instability, leading to increased ice discharge, thinning ice, and retreating grounding lines (Weertman, 1974; Schoof, 2007; Favier et al., 2014; Joughin et al., 2014). In parallel, increased surface air temperature can lead to surface melting, subsequent hydrofracturing, and possibly to major thinning and retreat of outlet glaciers after the collapse of ice shelves (DeConto and Pollard, 2016). Surface melting, leading to meltwater ponding, drainage into crevasses and hydrofracturing, is thought to be the main cause of the Larsen ice shelves collapse over the last decades in the Antarctic Peninsula (van den Broeke, 2005; Scambos et al., 2009; Vaughan et al., 2003). While surface melting is currently limited to relatively rare events over the Amundsen Sea ice shelves (Nicolas and Bromwich, 2010; Trusel et al., 2012) and underlying reasons for melt ponds formation versus active surface drainage network remains unclear (Bell et al., 2018), the rapid surface air warming observed (Steig et al., 2009; Bromwich et al., 2013) and projected (Bracegirdle et al., 2008) in this region suggests that surface melting could increase in the future. Our study focuses on the two atmospheric-related aspects that can significantly affect the contribution of the Amundsen Sea sector to sea level rise, i.e., snowfall accumulation that is expected to increase in a warmer climate and therefore to reduce the mean sea level, and surface melting that could potentially induce more ice discharge and therefore increase the mean sea level.

Understanding the interannual variability of SMB and surface melting is key to (i) quantify the signal to noise ratio in climate trends, (ii) identify opportunities for seasonal predictions, and (iii) assess the capacity of climate models to respond to global climate variability. Furthermore, years with particularly strong surface (or oceanic) melting could trigger irreversible grounding line retreat without the need for a long-term climate trend. Interannual variability in the Amundsen Sea region is usually described in

terms of connections with the El Niño Southern Oscillation (ENSO), the Southern Annular Mode (SAM), and the Amundsen Sea Low (ASL). Our study revisits these connections through dedicated regional simulations based on the MAR model (Fettweis et al., 2017; Agosta et al., 2019). Hereafter, we start by reviewing recent literature on these climate connections.

The El Niño Southern Oscillation (ENSO ; Philander et al., 1989) is the leading mode of ocean and atmosphere variability in the tropical Pacific. It is the strongest climate fluctuation at the interannual timescale and can bring seasonal to multi-year climate predictability (e.g. Izumo et al., 2010). Global climate models predict an increasing number of extreme El Niño events in the future, with large global impacts (Cai et al., 2014, 2017). Interannual and decadal variability in the tropical Pacific affects air

temperature (Ding et al., 2011), snowfall (Bromwich et al., 2000; Cullather et al., 1996; Genthon and Cosme, 2003), sea ice extent (Pope et al., 2017; Raphael and Hobbs, 2014) and upwelling of circumpolar deep water favoring ice-shelf basal melting (Dutrieux et al., 2014; Steig et al., 2012; Thoma et al., 2008) in West Antarctica. Recent studies found concurrences between El Niño events and summer surface melting over West Antarctic ice shelves (Deb et al., 2018; Nicolas et al., 2017; Scott et al., 2019). These

connections are generally explained in terms of Rossby wave trains excited by tropical convection during El Niño events and inducing an anticyclonic anomaly over the Amundsen Sea (Ding et al. 2011). Paolo et al. (2018) reported a positive correlation between ENSO and the satellite-based ice-shelf surface height in the Amundsen Sea over 1994-2017. Based on a detailed study of the extreme El Niño/La Niña sequence from 1997 to 1999, these authors suggested that El Niño events could increase snow accumulation but,

also increase ocean melting even more, thus leading to an overall ice shelf mass loss. The impact of ENSO was found to be stronger for the Dotson ice-shelf and eastward, and weaker for Pine Island and Thwaites (Paolo et al., 2018). However, the aforementioned studies were based on the analysis of few recent ENSO events, and did not account for the highly-variable properties of ENSO over multi-decadal periods (e.g. Deser et al., 2012; Newman et al., 2011).

The Southern Annular Mode (SAM; Hartmann and Lo, 1998; Limpasuvan and Hartmann, 1999; Thompson and Wallace, 2000) is the dominant mode of atmospheric variability in the Southern hemisphere, and corresponds to a variation of the strength and position of the circumpolar westerlies. Over the last three to five decades, the SAM has exhibited a positive trend, i.e., westerly winds have been strengthening and shifting poleward (Chen and Held, 2007; Jones et al., 2016; Marshall, 2003). Medley

and Thomas (2019) found similar patterns for the SAM trends and the reconstructed snow accumulation trend over 1801-2000. By contrast, the temperatures above the melting point over the Amundsen ice shelves were found to be largely insensitive to the polarity of the SAM (Deb et al., 2018). The SAM phase has also been suggested to influence the ENSO teleconnection to the south Pacific: in-phase ENSO and SAM events (i.e. El Niño/SAM- or La Niña/SAM+) favor anomalous transient eddy momentum fluxes

in the Pacific that make the ENSO teleconnection to the South Pacific stronger than average (Fogt et al., 2011).

The Amundsen sea low (ASL ; Raphael et al., 2016; Turner et al., 2013a) is a dynamic low-pressure system located in the Pacific sector of the Southern Ocean and moving across the Ross, Amundsen, and Bellingshausen seas. The ASL is important regionally and variations of its central pressure and position respectively reflect the second and third leading modes of the Southern hemisphere climate respectively (Scott et al. 2019 their figure 3). A westward shift of the ASL induces northerly flow anomalies over the Amundsen Sea, leading to warmer conditions and increased moisture transport over the ice sheet (Hosking et al., 2013; Thomas et al., 2015; Hosking et al., 2016; Raphael et al., 2016; Fyke et al., 2017). Variations in the ASL central pressure also largely impact the West Antarctic climate: anti-cyclonic anomalies near 120°W lead to marine air intrusion over the ice sheet, thereby increasing cloud cover, longwave downward radiations and therefore surface air temperature over WAIS (Scott et al., 2019). While a deepening of the ASL is predicted for the twenty-first century in response to greenhouse gas emissions, its high regional variability makes future changes of the ASL difficult to predict (Hosking et al., 2016; Turner et al., 2009).

Importantly, ENSO and SAM are not independent from each other and both modes of climate variability impact the ASL (Fogt and Wovrosh, 2015). SAM influences the ASL central pressure since it affects the mean sea level pressure over Antarctica (Turner et al., 2013a). The second and third leading modes of variability in the South Pacific have been suggested to be affected by Rossby wave trains induced by tropical convection anomalies (Mo and Higgins, 1998). In terms of ASL, it corresponds to a migration further west (east) during the La Niña (El Niño), but the difference has a low statistical significance (Turner et al., 2013b). Scott et al. (2019) recently reported that El Niño conditions favored blocking in the Amundsen Sea as well as a negative SAM phase, both leading to warm surface air anomalies in West Antarctica.

In this study we revisit the influence of ENSO, SAM and ASL on summer SMB and melting over the drainage basins of the Amundsen sector in West Antarctica for the 1979-2017 period. While the summer focus on melt rates is obvious, SMB in DJF (i.e. December-January-February) only represents 15% of the annual SMB. It is nonetheless interesting to analyze the similarities and differences in what drives SMB and melting, and the modes of variability and their teleconnections to the Amundsen Sea region both have strong seasonal characteristics, so that each season needs to be considered separately. To do so, we simulate the surface conditions of the Amundsen Sea region over 1979-2017 using the polar-adapted regional atmospheric model MAR forced by the ERA-Interim reanalysis. Section 2 describes the methodology followed in the study and presents the model and observations used for comparison. The model results are analyzed and evaluated against observations in Section 3, after evaluating the model skills (section 3.1), we analyze and discuss our results on the potential impact of large-scale climate variabilities on the SMB and melting in section 3.2 and 4. The conclusions are provided in Section 5.

## 2 Material and Method

### 2.1 Model

To estimate SMB and surface melt over the Amundsen sector we use the regional atmospheric model MAR (Gallée and Schayes, 1994) and specifically the version 3.9.3 (http://mar.cnrs.fr, last access: 25 September 2019). The model solves the primitive equations under the hydrostatic approximation. It solves conservation equations for specific humidity, cloud droplets, rain drops, cloud ice crystals and snow particles (Gallée, 1995; Gallée and Gorodetskaya, 2010). MAR represents coupled interactions between the atmospheric surface boundary layer and the snowpack using the Soil Ice Snow Vegetation Atmosphere Transfer (SISVAT) originally developed by De Ridder and Gallée (1998). The snow–ice part of SISVAT includes submodules for surface albedo, meltwater percolation and refreezing, and snow metamorphism based on an early version of the CROCUS model (Brun et al., 1992). MAR has been largely evaluated in polar regions (e.g. Amory et al., 2015; Gallée et al., 2015; Lang et al., 2015; Fettweis et al., 2017; Kittel et al., 2018; Agosta et al., 2019; Datta et al., 2019).

Our domain includes the drainage basins of the Amundsen Sea Embayment glaciers and a large part of the Amundsen Sea until 65°S using oblique stereographic projection (EPSG: 3031). It covers an area of 2800 x 2400 km at 10 km horizontal resolution (Fig.1) and 24 vertical sigma levels located from approximately 1 m to 15500 m above the ground. We use 30 snow layers, resolving the 20 first meters of the snowpack, with a fine vertical resolution at the surface (1 mm) increasing with depth, snow layer thickness varies dynamically depending on the physical properties of overlying snow layer properties. If neighboring layers get similar properties then layers are associated together. The radiative scheme and cloud properties are the same as in Datta et al. (2019) and the surface scheme including snow density and roughness are the same as in Agosta et al. (2019). The model is forced, over the period 1979-2017, by ERA-interim reanalysis (Dee et al., 2011), which performs well over Antarctica (Bromwich et al., 2011; Huai et al., 2019), at 6 hourly temporal resolution and relaxed over ~50km laterally (pressure, wind, temperature, specific humidity, the relaxation zone is shown in Fig.1), at the top (i.e. above 10 km) of the troposphere (temperature, wind), and at the surface (sea ice concentration, sea surface temperature). The Bedmap2 surface elevation dataset is used for the ice-sheet topography (Fretwell et al., 2013). The snowpack density and temperature are initialized from the pan-Antarctic simulation from Agosta et al. (2019). Drifting snow is relatively infrequent in the Amundsen region (Lenaerts et al., 2012) so that the drifting snow module has been switched off in our configuration, similarly as in Agosta et al. (2019).

In section 3.2 we provide the SMB constituents averaged over individual drainage basins.

## 2.2 Antarctic surface observations

We make use of meteorological data from the SCAR database including observations from the Italian Antarctic Research Program (http://www.climantartide.it, last access: 25 September 2019), the Antarctic Meteorological Research Center (AMRC program) (http://amrc.ssec.wisc.edu/, last access: 25 September 2019) and the Australian Antarctic automatic weather station (AWS) dataset (http://aws.acecrc.org.au/, last access: 25 September 2019). Among the 243 AWS available over Antarctica since 1980, we selected

the 41 stations (see Table S1 for station names) located no more than 15 km from the closest continental MAR grid point (even if the domain resolution is 10 km, stations over islands or capes that are not resolved can be located farther than 15 km from the closest continental MAR grid point). For each location, modelled values (surface pressure, near-surface temperature and near-surface wind speed) are computed as the average-distance-weighted value of the four nearest continental grid points. A second

selection criterion is also applied in order to reduce comparison errors due to the difference between the model surface elevation and the actual AWS elevation: we only retain observations with an elevation difference lower than 250 m. This two-stage selection leaves 41 suitable AWS in our domain (Fig. 1).

      To evaluate the simulated SMB, we use airborne-radar data from Medley et al. (2013, 2014) covering the period 1980-2011. These data were collected through the NASA's Operation Icebridge campaign over

the Thwaites and Pine Island basins. They are based on the CReSIS radar (Center for Remote Sensing of Ice Sheets), which is an ultra-wideband radar system able to measure the stratigraphy of the upper 20 – 30 m of the snowpack with few centimeters in vertical resolution. Airborne-radar data were verified with 190 firn cores accumulation records. To evaluate the SMB regional pattern at a broader scale, we also compared the simulated SMB with the observations gathered in the GLACIOCLIM-SAMBA dataset

thoroughly described by Favier et al., (2013) and updated by Wang et al., (2016) that are covered by our domain. Similarly, to Kittel et al., (2018) and Agosta et al., (2019), we selected the observations whose the measurement period extends from 1950 to 2018. Observations before 1979 (i.e., the beginning of our study period), were compared to the average SMB simulated by MAR provided they cover a period of at least 5 years, while observations after 1979 were compared to the SMB modelled by MAR for the

observation period. We then compared the modelled SMB computed by using a four-nearest inverse-distance-weighted method to each of the 124 selected SMB observations.

      To evaluate simulated surface melt, we use satellite-derived estimates of surface meltwater production over 1999-2009 from Trusel et al., (2013), provided at 4.45 km resolution, and based on the QuickSCAT backscatter and calibrated with in-situ observations. We also use data from Nicolas et al. (2017) who

provide the number of melt days at 25 km resolution over Antarctica. This product is based on passive microwave observations from the Scanning Microwave Multichannel Radiometer (SMMR), the Special Sensor Microwave/Imager (SSM/I), and the Special Sensor Microwave Imager Sounder (SSMIS) spaceborne sensors, and covers the 1978-2017 period. For a given grid cell and a given day, melt is

assumed to occur as soon as one of the two daily observations of brightness temperature exceeds a
threshold value. As the identification of melt days may be sensitive to the algorithm, we also use the
dataset from Picard et al. (2007), extended to 2018 (http://pp.ige-grenoble.fr/pageperso/picardgh/melting/, last access: 25 September 2019). This dataset is also based on
SMMR and SSMI, but uses the algorithms from Torinesi et al. (2003) and Picard and Fily (2006) to
retrieve melt days. It is provided as daily melt status at 25km resolution over the Antarctic continent from
1979 to 2018.

## 2.3 Climate indices

To describe the ENSO, we use the Southern Oscillation Index (SOI) index from the Global Climate
Observing System (GCOS) Working Group on Surface Pressure (Ropelewski and Jones, 1987;
https://www.esrl.noaa.gov/psd/gcos_wgsp/Timeseries/SOI/, last access: 25 September 2019). The SOI
index corresponds to the normalized pressure difference between Tahiti and Darwin based on
observations. The Rossby wave trains connecting the Equatorial Pacific to Antarctica are expected to
develop within a few weeks in response to ENSO anomalies (e.g. Hoskins and Karoly, 1981; Mo and
Higgins, 1998; Peters and Vargin, 2015), so we first use the synchronous (DJF) SOI index in section 3.
The lagged relationship to ENSO is discussed in section 4, where we use other 3-month averages of SOI
such as JJA (June-July-August). SOI is preferred to NINO3.4 because it gives slightly stronger
correlations with the variability in the Amundsen Sea region (as also found by Scott et al., 2019 and
Holland et al., 2019) but very similar results were obtained using NINO3.4 (not shown).

We use the SAM index from NOAA/CPC (https://stateoftheocean.osmc.noaa.gov/atm/sam.php,
last access: 25 September 2019) to describe the primary mode of atmospheric variability in the Southern
Ocean (e.g., Marshall, 2003). The SAM index is calculated as the difference of mean zonal pressure
between the latitudes of 40°S and 65°S based on NCEP-NCAR reanalysis which produces a SAM that is
consistent with other reanalyses after 1979 (Gerber and Martineau, 2018). In the negative (positive) phase,
the mean sea level pressure anomaly between the Antarctic and mid latitude is positive (negative) and
leads to a weaker (stronger) polar jet. Thus, positive (negative) values of the SAM index correspond to
stronger (weaker)-than-average westerlies over the mid-high latitudes (50°S-70°S) and weaker (stronger)
westerlies in the mid-latitudes (30°S-50°S).

We use two other indices to describe the evolution of the migration and intensity variations of the
Amundsen Sea Low (ASL). The datasets are provided by the British Antarctic Survey
(https://legacy.bas.ac.uk/data/absl/ASL-index-Version2-Seasonal-ERA-Interim_Hosking2016.txt, last
access: 25 September 2019), and calculated from the ERA-Interim reanalysis. To describe the migration,
we use the longitudinal position of the ASL defined as the position of the minimum pressure within the
box 170°-298°E, 80°-60°S (Hosking et al., 2016), defined in degree East. A decrease in the longitudinal
position index hence corresponds to a westward shift of the ASL. To describe the intensity of the ASL,

we use the relative central pressure of the ASL calculated as the minimum pressure in the aforementioned box minus the average pressure over that box (Hosking et al. 2016). A more intense ASL (deeper depression) is therefore represented by a lower index.

The SAM and ASL indices are defined regionally, and we do not expect any lag with summer SMB, so these indices are therefore calculated as DJF averages. All the correlations are calculated using detrended time series.

The correlations between these four indices are indicated in Table 1. A significant anti-correlation is obtained between the SAM index and ENSO (i.e. -SOI) as previously reported by Fogt et al. (2011). There is no significant relationship between the ASL longitudinal position and ENSO or SAM, as previously reported by Turner et al., (2013a). The relative central pressure also varies independently from SAM, ENSO, and the ASL longitudinal position. Numerous previous studies used the absolute rather than relative central pressure to characterize the ASL, but this index is strongly correlated to the SAM index and cannot be considered independently (Table 1). As proposed by Hosking et al. (2013), the ASL relative central pressure (i.e actual central pressure minus pressure over the AS sector) allows for a better understanding of West Antarctic climate as it removes the influence of large scale variability such as ENSO and SAM.

Table 1: Correlation between climate indices (-SOI, SAM, ASL longitudinal position, ASL relative central pressure, ASL actual central pressure) in austral summer (DJF). Values in brackets represents the percentage of significance.

| Statistical correlation (R) | -SOI | SAM | ASL Longitudinal position (°East) | ASL relative central pressure (hPa) | ASL actual central pressure (hPa) |
|---|---|---|---|---|---|
| -SOI | | -0.45 (99%) | -0.22 (82%) | 0.00 (1%) | 0.40 (99%) |
| SAM | | | 0.18 (73%) | -0.25 (88%) | -0.88 (99%) |
| ASL Longitudinal position (°East) | | | | -0.23(84%) | -0.15 (63%) |


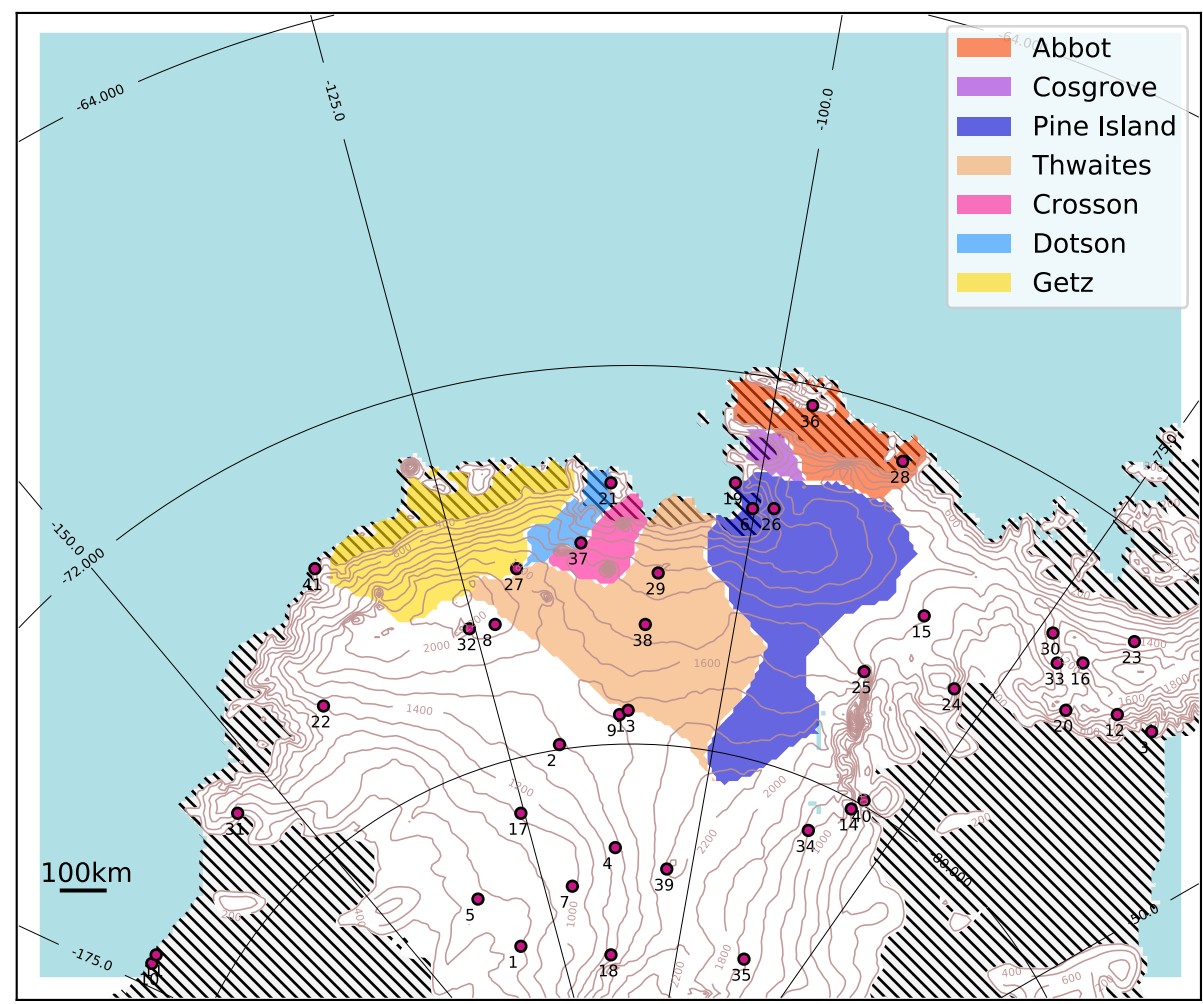

**Figure 1 : Simulation domain. The drainage basins (Rignot et al., 2019) under consideration in this paper are shaded in color and Automatic Weather Station (AWS) are indicated with red point. Hatched area represents ice-shelves and contour line the surface elevation (every 200m). Station name from 1 to 41 : (1) Brianna, (2) Byrd, (3) Cape Adams, (4) Doug, (5) Elizabeth, (6) Evans Knoll, (7) Harry, (8) Janet , (9) Kominko-Slade, (10) Martha2, (11) MarthaI, (12) Mount McKibben, (13) Noel, (14) Patriot Hills, (15) Siple Dome, (16) Ski Hi, (17) Swithinbank, (18) Theresa, (19) Backer Island, (20) Bean Peaks, (21) Bear Peninsula, (22) Clarke Mountains, (23) Gomez Nunatak, (24) Haag Nunatak, (25) Howard Nunatak, (26) Inman Nunatak, (27) Kohler Glacier, (28) Lepley Nunatak, (29) Lower Thwaites Glacier, (30) Lyon Nunatak, (31) Mount Paterson, (32)  Mount Sidley, (33) Mount Suggs, (34) Pirrot Hills, (35) Steward Hills, (36) Thurston Island, (37) Toney Mountain, (38) Up Thwaites Glacier, (39) Whitmore Mountains, (40) Wilson Nunatak, (41) Russkaya. Relaxation zone is shown in white (~50km).**


## 3 Results

We first evaluate the simulations with regard to observations (section 3.1). Then, we analyze the interannual variations in SMB and melting (section 3.2).

### 3.1 Model evaluation

We first evaluate the near-surface temperature and near-surface wind speed in comparison to AWS data (Fig.2).

Our MAR configuration reproduces the daily near-surface temperatures, with a mean bias of 0.10 °C and a mean correlation of 0.93 for the whole year and 0.86 for summer months (Fig.2a). The statistics per station show a Root Mean Square Error (RMSE) varying from 2.66 (10[th] percentile) to 4.15 °C (90[th] percentile) and a mean bias varying from -1.97 to 1.31 °C for the whole year (see supplementary material

for more details).

The model tends to overestimate the lowest observed wind and underestimate the highest observed wind speeds (regressions in Fig.2b). The model agreement with observations is nonetheless good on average, with a mean underestimation of 0.42 m s⁻¹. The statistics per station show a RMSE varying from 1.73 to 3.69 m s⁻¹, and a mean bias varying from -3.08 to 0.85 m s⁻¹ for the whole year. The variance of the wind speed simulated by MAR is lower than observed. Less satisfactory results are generally found for the stations located on an island. This can be explained by the resolution of 10 km which is still too coarse to resolve small topographic features. For both, near-surface temperature and wind speed, the statistics for the summer period (DJF) are very similar to the statistics for the whole year. Our results show very similar model skills compared to other simulations in the same region (Deb et al., 2016; Lenaerts et al., 2017) or at coarser resolution over the whole ice sheet (Agosta et al., 2019).

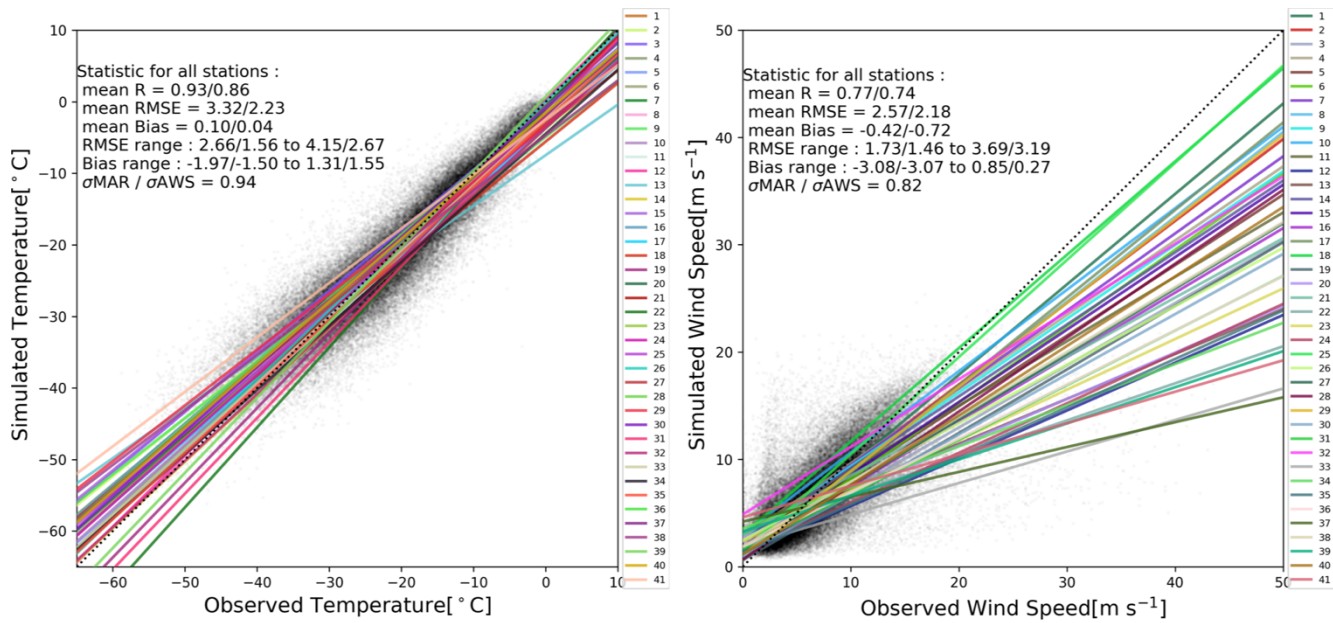

**Figure 2 :** Scatter plots of observed vs. simulated daily near-surface temperature (a) and daily near-surface wind speed (b) for the selected AWS (see corresponding locations and names in Fig. 1). The statistics, including RMSE, correlation (R), bias, and standard deviations (σ), are calculated for individual stations and provided as multi-station mean over the whole year / over the summer months (DJF). The range of RMSE and biases across individual stations is also indicated with the 10th percentile and the 90th percentile of all RMSE values. The lines represent least-mean-square linear fit between simulated data and observations. The complete statistical analyses for individual AWS are provided in Supplementary material (Table S1-S2).

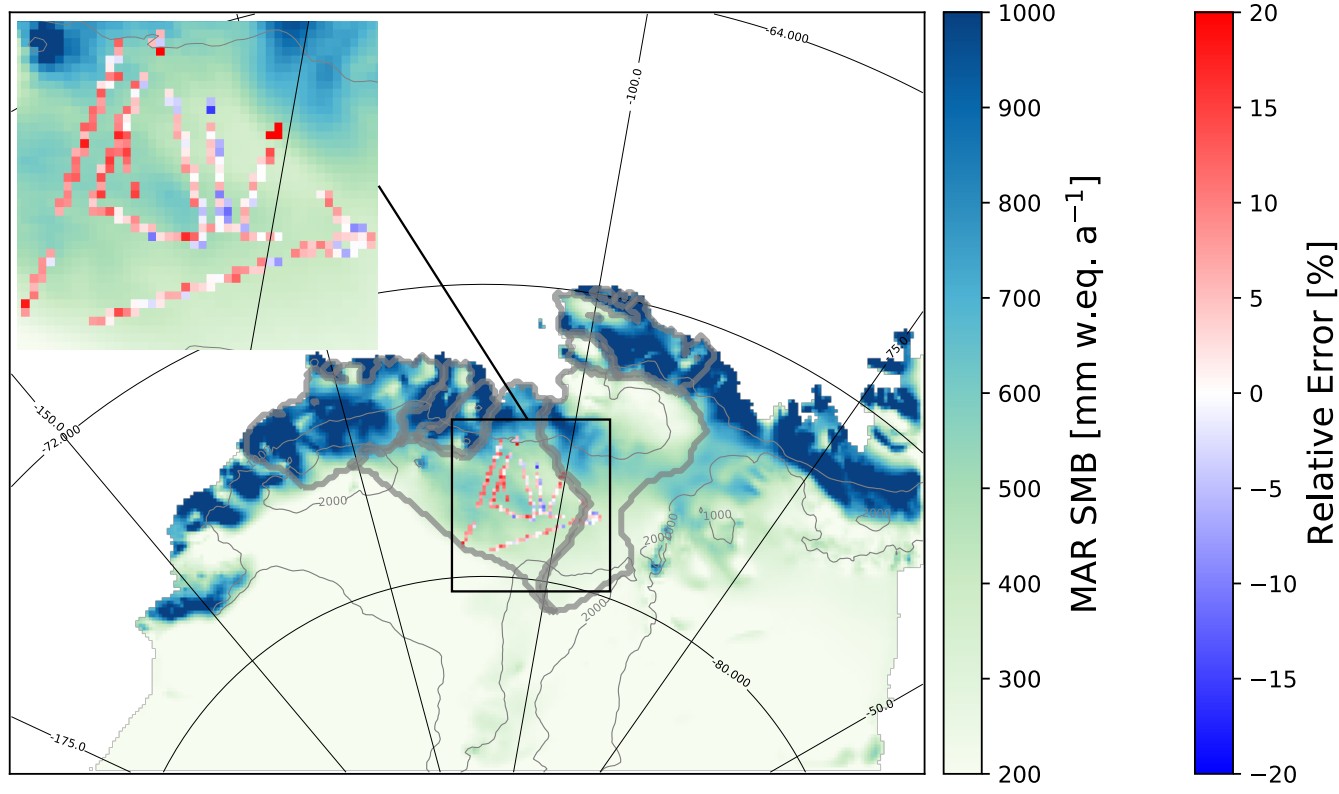

**Figure 3 : Annual mean (1979-2017) simulated SMB (blue-green scale) and relative error of the simulated SMB compared to the airborne-radar data from Medley et al. (2013, 2014) (blue-red color bar). Grey contours indicate the surface height (every 1000m). The drainage basins under consideration are the same as in Fig.1 (large grey contours here).**

We now assess the simulated SMB compared to the SMB from Medley et al., (2013, 2014) derived from airborne radar over the period 1980-2011. The simulated SMB is well captured by MAR with a mean relative overestimation of approximately 10% over the Thwaites basin, and local errors smaller than 20% at all locations (Fig.3). The interannual variability is also well simulated by MAR with a correlation of 0.90 (Fig.4). In order to have a broad overview of the SMB evaluation, we also compared the simulated SMB with the GLACIOCLIM-SAMBA dataset (Favier et al., 2013) over the Ross and Siple Coast sector (See Fig.S1 in Supplementary material). The bias of simulated SMB compared to observation is less than 10 mm w.e $a^{-1}$ and local bias can reach 30 mm w.e $a^{-1}$. However, the relative bias between GLACIOCLIM-SAMBA dataset and simulated SMB is more pronounced with only 44% of GLACIOCLIM-SAMBA sites show a relative error with simulated SMB lower than 20%. All SMB components are shown in Table 2.

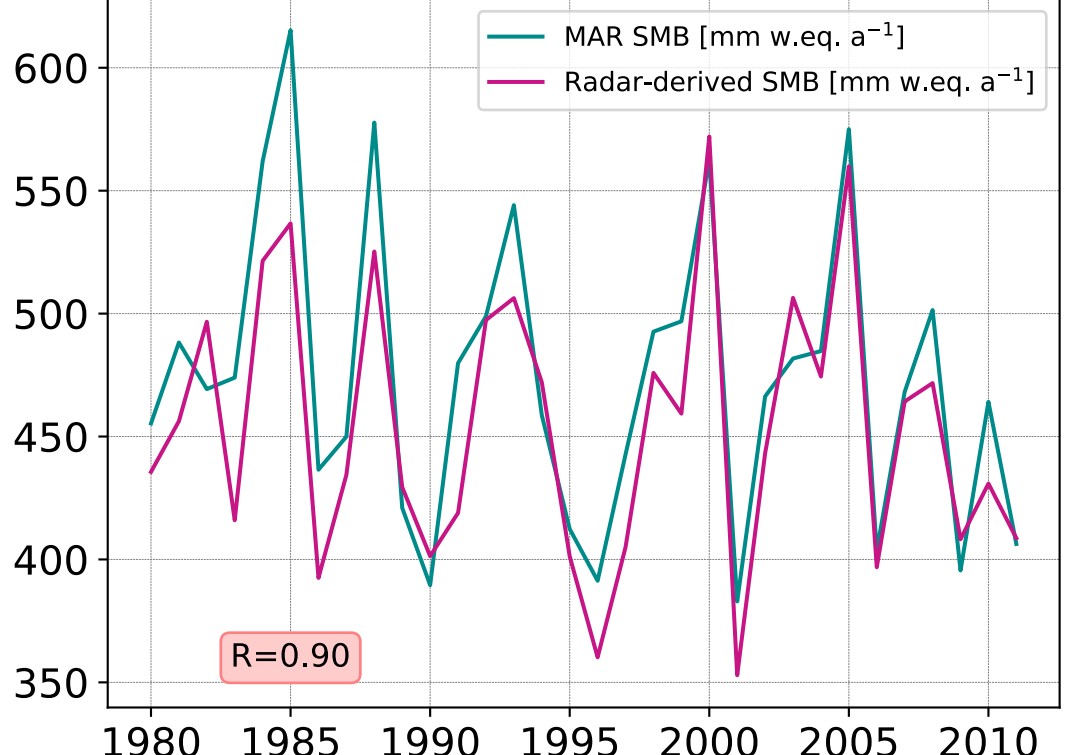

**Figure 4 : Timeseries of the annual mean (January to December) simulated and radar-derived SMB from 1980 to 2011 over the Thwaites basins.**

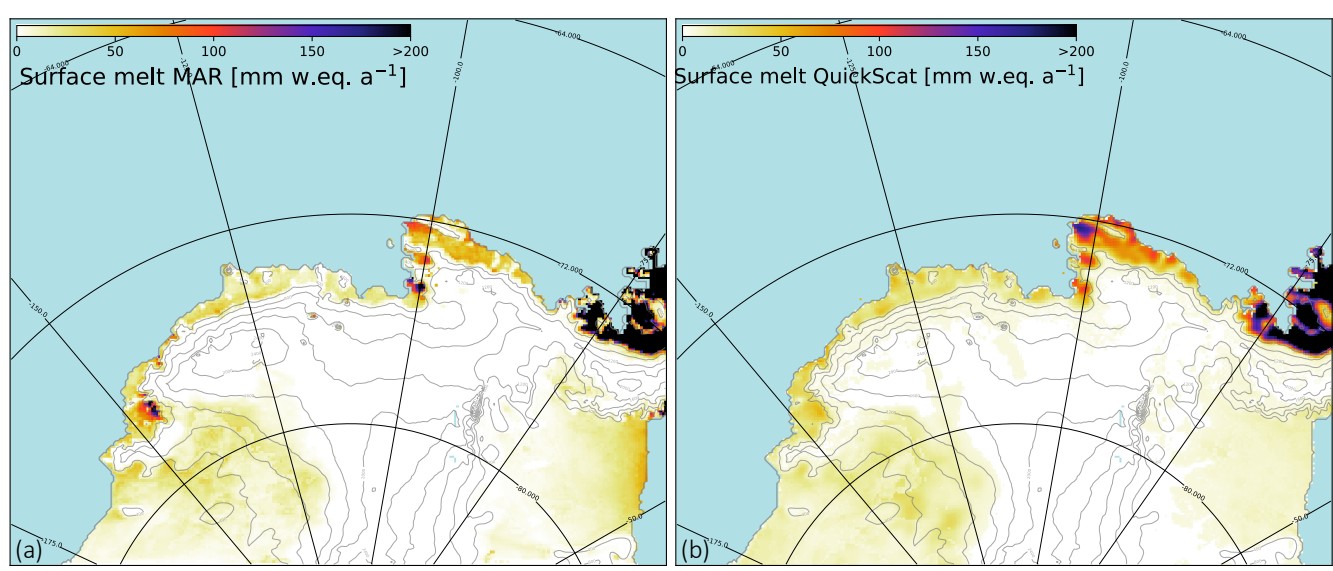

**Figure 5 : Annual surface melt rate (a) simulated by MAR over 1999-2009, and (b) derived from QuickScat satellite data over the same period (Trusel et al. 2013) and interpolated over the MAR grid.**

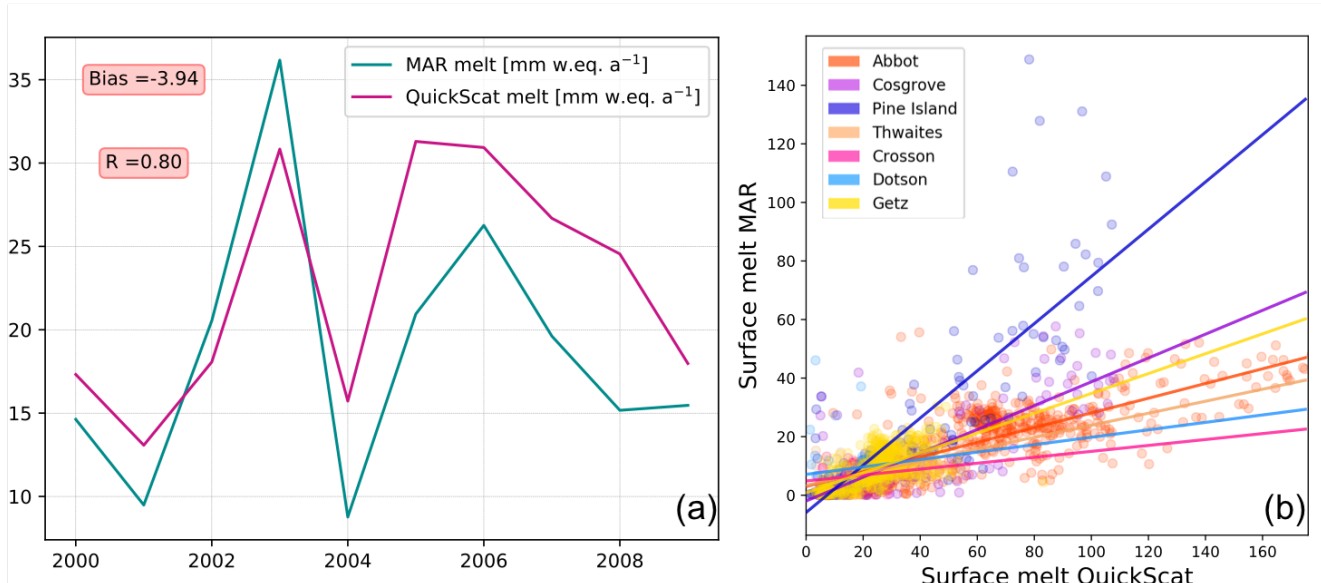

**Figure 6 : (a) Time series of surface melt rates in mean over the model domain derived from satellite data and simulated by MAR, years labelled on the X-axis refer to the second year of a given austral summer (e.g., summer 1999–2000 is labelled 2000). and (b) Surface melt modelled versus surface melt interpolated from satellite data (QuickScat) over drainage basins, (only where surface melt > 0 mm w.e. a$^{-1}$) and over the period 1999-2009.**

The areas of highest surface melt (> 100 mm w.e. a$^{-1}$) are located near the coast and particularly over Abbot, Cosgrove, and the eastern part of Pine Island ice shelf, while more extreme values (>200 mm w.e. a$^{-1}$) are found near the Peninsula in both simulated and observed datasets (Fig. 5). Even if the simulated and observed patterns are similar, the simulated surface melt is a factor of two lower than observations locally (e.g. over Abbot ice shelf and the Peninsula). While the interannual melt rate variability is well reproduced with a correlation of 0.80, the surface melt rate simulated by MAR is underestimated by 18% on average compared to QuickScat estimates (Fig.6a). Surface melt rate over Pine Island basins is well simulated by MAR (Fig.6b) with R equal to 0.80 compared to drainage basins with low surface melt (i.e Crosson, Dotson) where R is equal to 0.14 and 0.24 respectively. This melt underestimation, particularly pronounced over drainage basins with low surface melt rate, could be explained by the slight overestimation of the snowfall accumulation (10-20%), as the presence of a fresh snow layer of high albedo overlying snow or ice layers of lower albedo likely reduces melt. MAR surface melt presents slight overestimation over Getz ice-shelf (Fig.5) possibly explained by wind advection, föhn effect, or even snow metamorphism simulated by MAR. Further work is needed to understand such local biases. MAR is fully driven by low resolution ERA-Interim sea ice cover and temperature therefore possible underestimation of the presence of polynyas can also play a role in the melt biases.

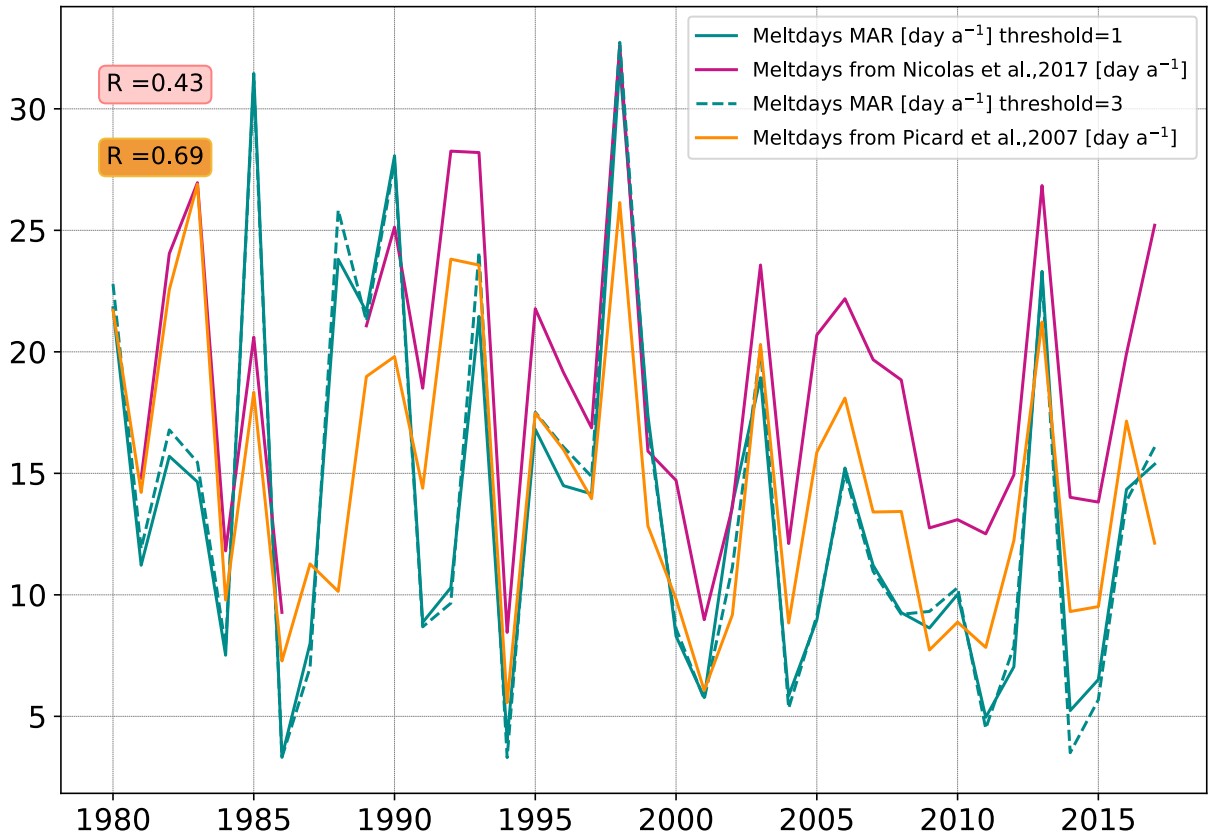

**Figure 7 : Time series of number of melt days per summer (DJF) averaged over the part of the domain with more than 3 melt days per year on average (which approximately corresponds to the ice shelf zone), derived from two satellite products and simulated by MAR (defined using a melt-rate threshold of either 1 or 3 mm w.e.day⁻¹).**

We also compare the number of melt days to the satellite products from Nicolas et al. (2017) and Picard et al. (2007). To avoid no melt days area in the timeseries computation we use the area where annual number of melt days for each dataset is more than 3 melt days per year, that corresponds approximately to the ice shelf zone. As with the amount of surface melt, the number of melt days over the domain is underestimated by MAR (Fig.7). The amplitude of the underestimation is not very sensitive to the melt-rate threshold used to define a melt day in MAR. A threshold of 1 mm w.e. day⁻¹ (as in Datta et al., 2019) gives a mean underestimation of 4.8 days per year compared to observation from Nicolas et al. (2017), while a threshold 3 mm w.e. day⁻¹ (as in Deb et al., 2018; Lenaerts et al., 2017) gives a mean underestimation of 4.9 days per year. This underestimation is less pronounced (0.8 to 0.9 day per year depending on the threshold) when using Picard et al. (2007) as a reference. The interannual variability in the number of melt days is reproduced with correlations of 0.69 and 0.43 to the two satellite products (Fig. 7). Previous study on Antarctic Peninsula also found that MAR melt occurrence is comparable to satellite products, but slightly underestimated over the Western coast of the Peninsula (Datta et al., 2019).

Overall, MAR well simulates the interannual variability of the Amundsen sector, and we are now going to use these simulations to investigate the drivers of interannual variability of SMB and surface melting.

### 3.2 Drivers of summer interannual variability

In this subsection, we first investigate the large-scale conditions leading to interannual anomalies in summer SMB or surface melting. For a sake of clarity, we only consider the Pine Island and Thwaites basin (together) as a first approach. To identify large-scale conditions leading to high (low) SMB, we calculate composites defined as the average of summers presenting a SMB greater than the 85$^{th}$ (lower than the 15$^{th}$) interannual percentile, and we proceed similarly for surface melt composites. We choose

the 85$^{th}$ and 15$^{th}$ percentiles to optimize the signal-to-noise ratio.

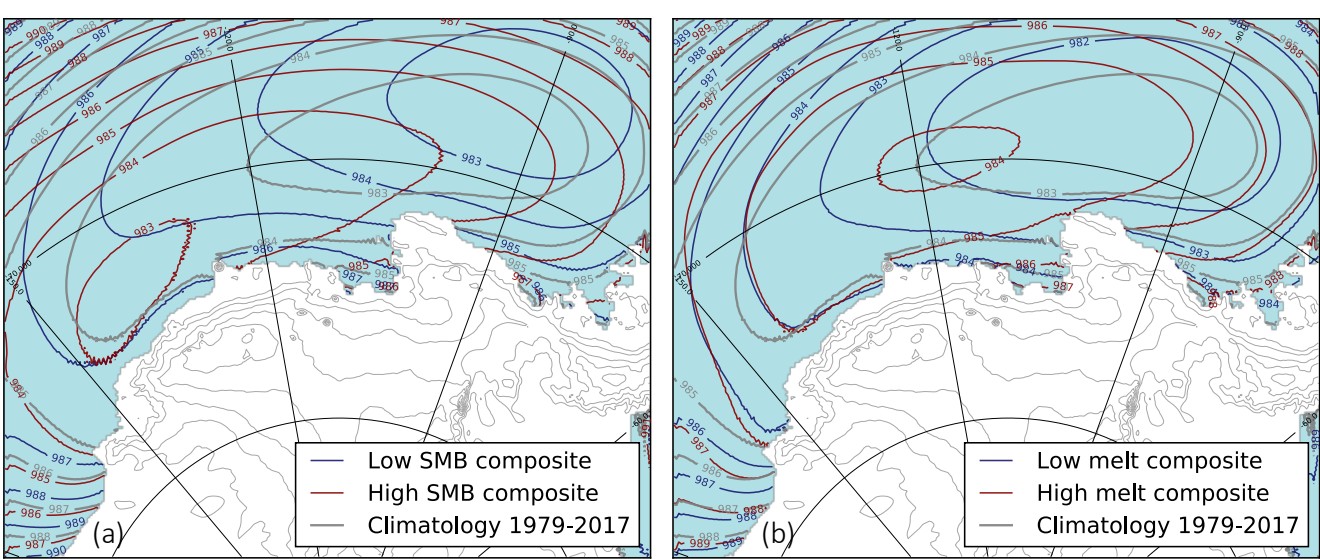

**Figure 8 : Summer sea surface pressure composites for high/low SMB (a) and high/low surface melt (b). The ice-sheet height is indicated by thin grey contours (every 500m).**

      Sea surface pressure composites show that distinct mechanisms affect the interannual variability of summer SMB and surface melting (Fig. 8). Summers with high SMB are on average characterized by a

far westward (by ~30°) and southward (by 3-4°) migration of the ASL center, while the reverse migration is found for summers with low SMB although with a smaller displacement (~15° eastward). In contrast, years with high surface melt rates are characterized with a much smaller ASL migration and no migration is found for years with low surface melt rates, but the pressure gradients differ between the high and low composites. Therefore, we hereafter consider the variability of SMB and surface melting separately.

To further characterize the tropospheric circulation associated with years of low or high summer SMB, we plot composites of both the 500 hPa geopotential height (Fig. 9a,b) and the 500 hPa geopotential height divided by the domain-averaged value for each season (Fig. 9c,d). The latter has the advantage to highlight changes in regional gradients (related to the regional circulation) rather than larger scale changes in geopotential height. Both provide similar composites, but the statistical significance is higher in

Fig. 9c,d.  On average, low-SMB summers are characterized by a northward and eastward ASL migration

(shown through a dipole in the 500hPa normalized geopotential composite in Fig. 9a,c), which is associated with an offshore surface wind anomaly over the glaciers of the Amundsen Sea sector (Fig. 9e). Conversely, high-SMB summers are characterized by a southward and westward ASL migration (Fig. 9b,d), which is associated with an onshore surface wind anomaly over the glaciers of the Amundsen

sector (Fig. 9f). The circulation anomalies typical of high-SMB summers favor the southward transport of precipitable water as indicated by the composites of integrated vapor transport (Fig.10a,b). Increased moisture transport towards the Amundsen Sea Embayment leads to denser cloud cover (Fig.10c,d) and increased SMB.

On average, high-melt summers are also associated with increased moisture transport towards the Amundsen Sea Embayment, and conversely for low-melt summers (Fig.11a,b), but the mechanism is somewhat different from the case of SMB. The ASL migration during high-melt summers is much smaller than for the high-SMB summers (Fig.8b). As previously done for SMB, we plot composites of both the 500 hPa geopotential height (Fig. 12a,b) and the 500 hPa geopotential height divided by the domain-

averaged value for each season (Fig. 12c,d), the latter better highlighting regional circulation changes (geopotential gradients). Summers with high surface melt rates show a significant increase in the 500 hPa geopotential height over the Bellingshausen Sea (Fig. 12b), i.e. an anticyclonic anomaly, and small westward ASL migration as shown in the 500hPa normalized geopotential composite (Fig. 12d). This anomaly is against the ASL mean circulation and creates a northerly flow anomaly over the ice sheet in

the Amundsen sector (Fig. 12e,f). This anticyclonic anomaly was described by Scott et al. (2019) in terms of enhanced blocking activity. As in Scott et al. (2019), we find that high-melt summers are associated with denser cloud cover (Fig.11c,d), increased downward longwave radiation (Fig.11e,f), and therefore surface air warming, while the opposite occurs for low-melt summers. Composites of sensible heat flux indicate that heat is lost by the snow surface to the atmosphere for high-melt summers, i.e. high melt

summers are not caused by föhn events on average (Fig. S2).

**Table 2 : Annual SMB decomposition for all drainage basins over 1979-2017 with SMB = Snowfall + Rainfall – Sublimation – Runoff. The middle rows indicate other terms that are not directly part of the SMB. The last two rows give snowfall and melt rates averaged over the ice shelves.**

| [mm.w.e yr$^{-1}$] | Abbot | Cosgrove | Pine Island | Thwaites | Crosson | Dotson | Getz |
|---|---|---|---|---|---|---|---|
| SMB | 959.5 | 660.5 | 429.1 | 504.5 | 867.7 | 895.0 | 843.0 |
| Sublimation | 26.5 | 30.3 | 12.7 | 0.6 | 22.6 | 25.6 | 22.8 |
| Snowfall | 981.9 | 688.5 | 441.3 | 505.0 | 887.6 | 919.5 | 864.9 |
| Rainfall | 4.0 | 2.3 | 0.4 | 0.1 | 2.8 | 1.1 | 0.8 |
| Runoff | 0.0 | 0.0 | 0.0 | 0.0 | 0.0 | 0.0 | 0.0 |
| Refreezing | 36.4 | 27.0 | 4.3 | 1.0 | 6.2 | 7.2 | 9.6 |
| Surface melt | 32.5 | 24.8 | 3.9 | 0.9 | 3.4 | 6.1 | 8.8 |
| Snowfall (only over ice shelf) | 795.4 | 296.9 | 422.7 | 811.5 | 1051.5 | 672.0 | 789.9 |
| Surface melt (only over ice shelf) | 57.9 | 83.2 | 82.0 | 26.5 | 18.5 | 23.7 | 26.7 |


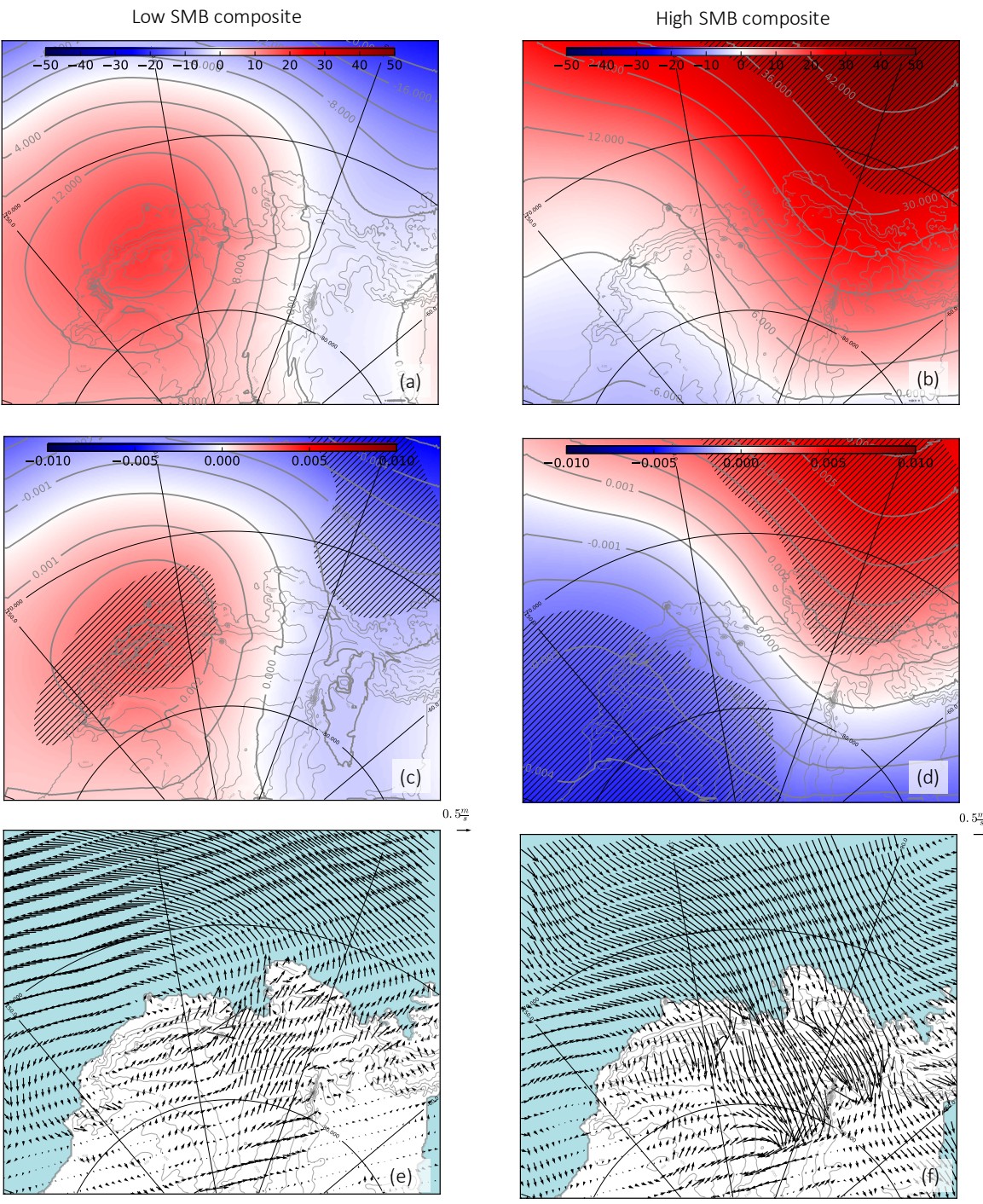

**Figure 9 : (a,b) 500 hPa geopotential height (m), (c,d) 500 hPa geopotential height divided by the domain-averaged value for each season and (e,f) 10m wind (m s⁻¹) anomalies during low-SMB summers (left) and high-SMB summers (right), scales of arrow lengths are shown near the upper right corner of panels (e) and (f). Anomalies are calculated as high/low composites minus the climatology over 1979-2017. Hatched area (a-d) represents significance>90% calculated with Welch's *t*-test.**


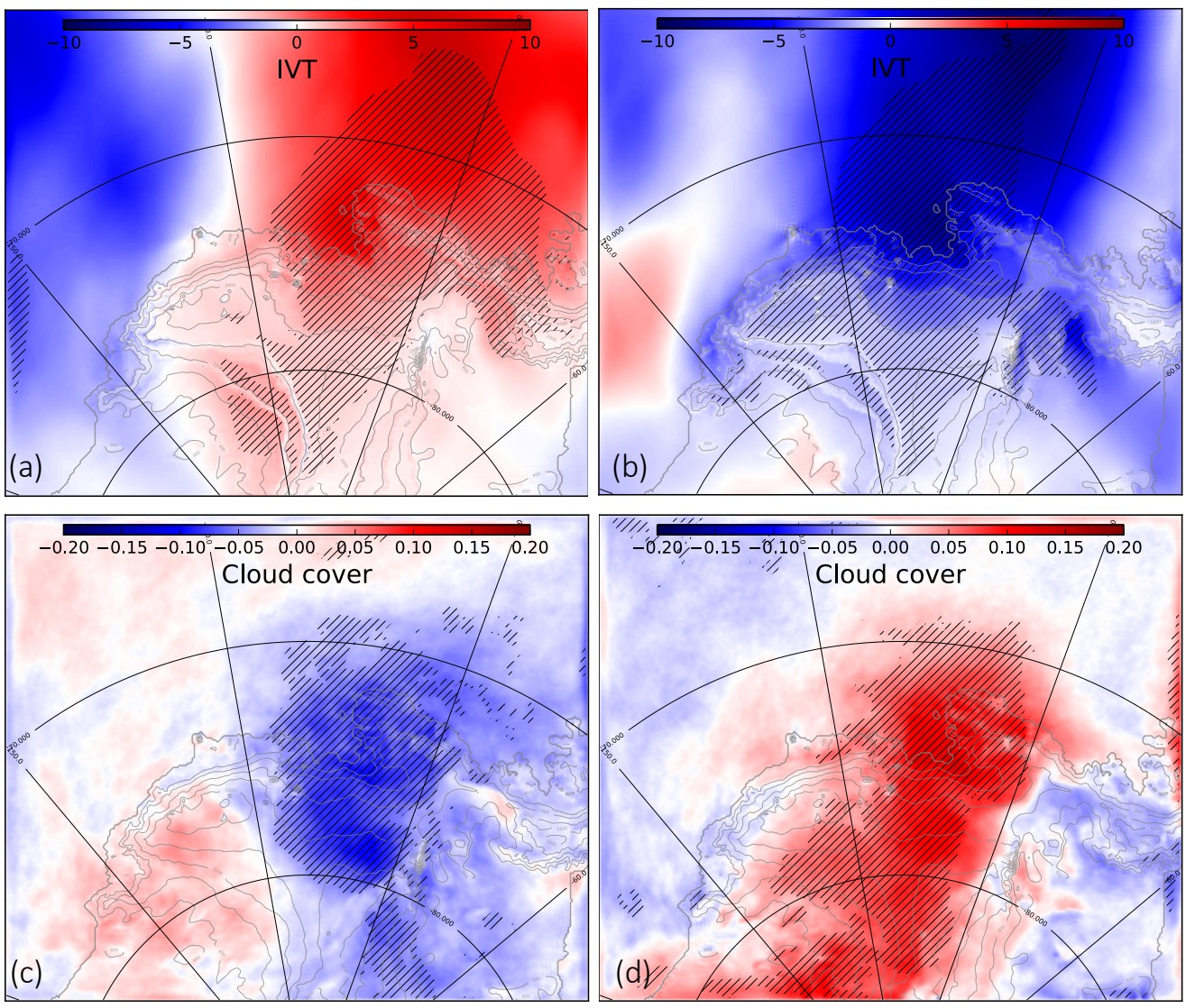

**Figure 10 :** **(a,b) vertical Integrated Vapor Transport (IVT) along the y-axis (negative toward the continent) calculated as**

$$IVT\ [kg\ m^{-2}] = \int_{925}^{700} q.v\ \frac{dP}{g}$$ **, with q the specific humidity (g kg⁻¹), v the wind speed (m s⁻¹), P the pressure (Pa), and g the gravity**

**(9.81 m s⁻²) and (c,d) cloud cover (no units, from 0 to 1) anomalies during low-SMB summers (left) and high-SMB summers (right).**

**Anomalies are calculated as high/low composites minus the climatology over 1979-2017. Hatched area represents significance>90% calculated with Welch's *t*-test.**

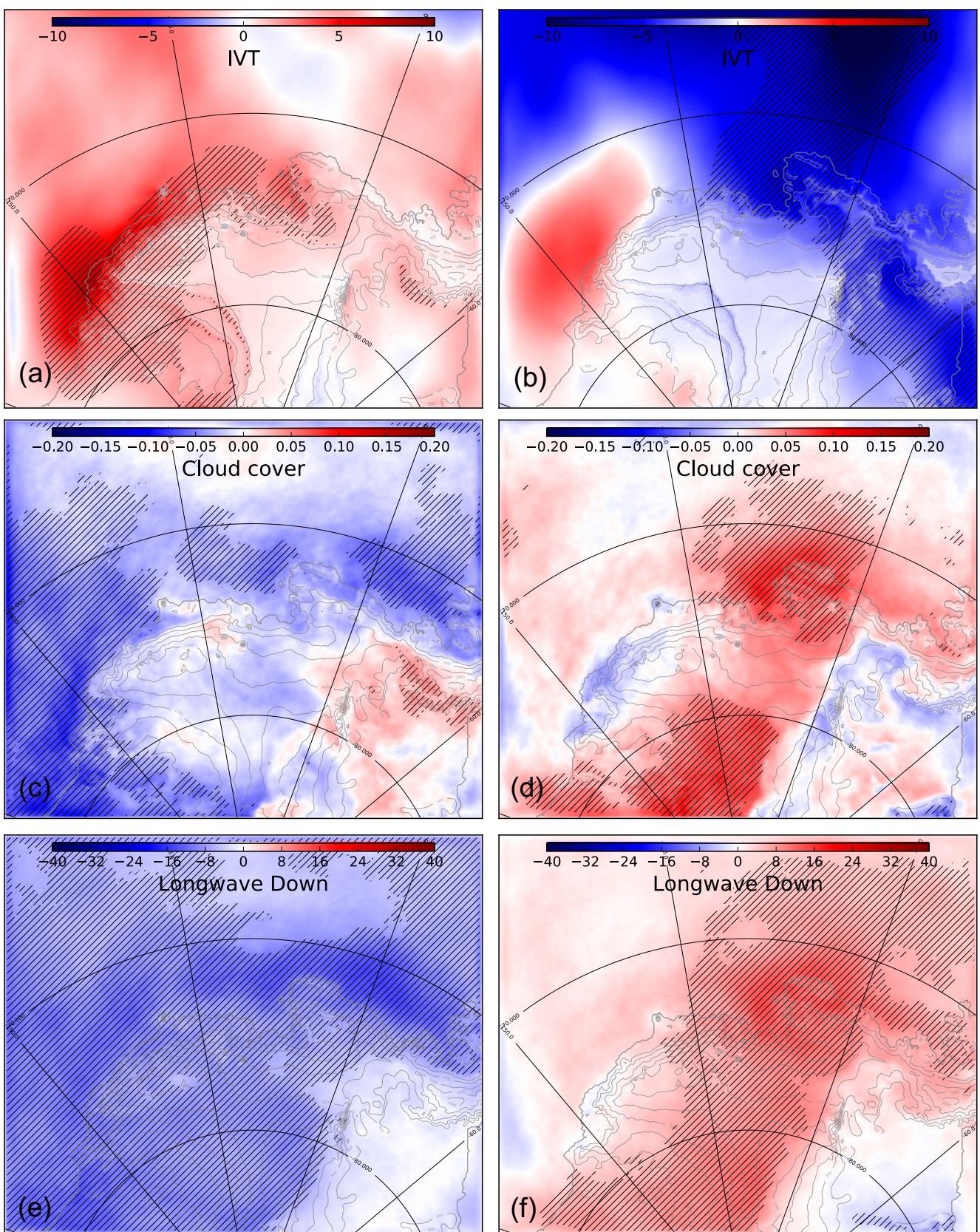

**Figure 11 :** (a,b) vertical Integrated Vapor Transport (IVT) along y axis (negative toward the continent (kg m$^{-2}$) (same formula as for Fig.10), (c,d) cloud cover (no units, from 0 to 1), and (e,f) downward longwave radiation (W m$^{-2}$) anomalies during low-melt summers (left) and high-melt summers (right). Anomalies are calculated as high/low composites minus the climatology over 1979-2017. Hatched area represents significance>90% calculated with Welch's *t*-test.

Low surface melt composite High surface melt composite

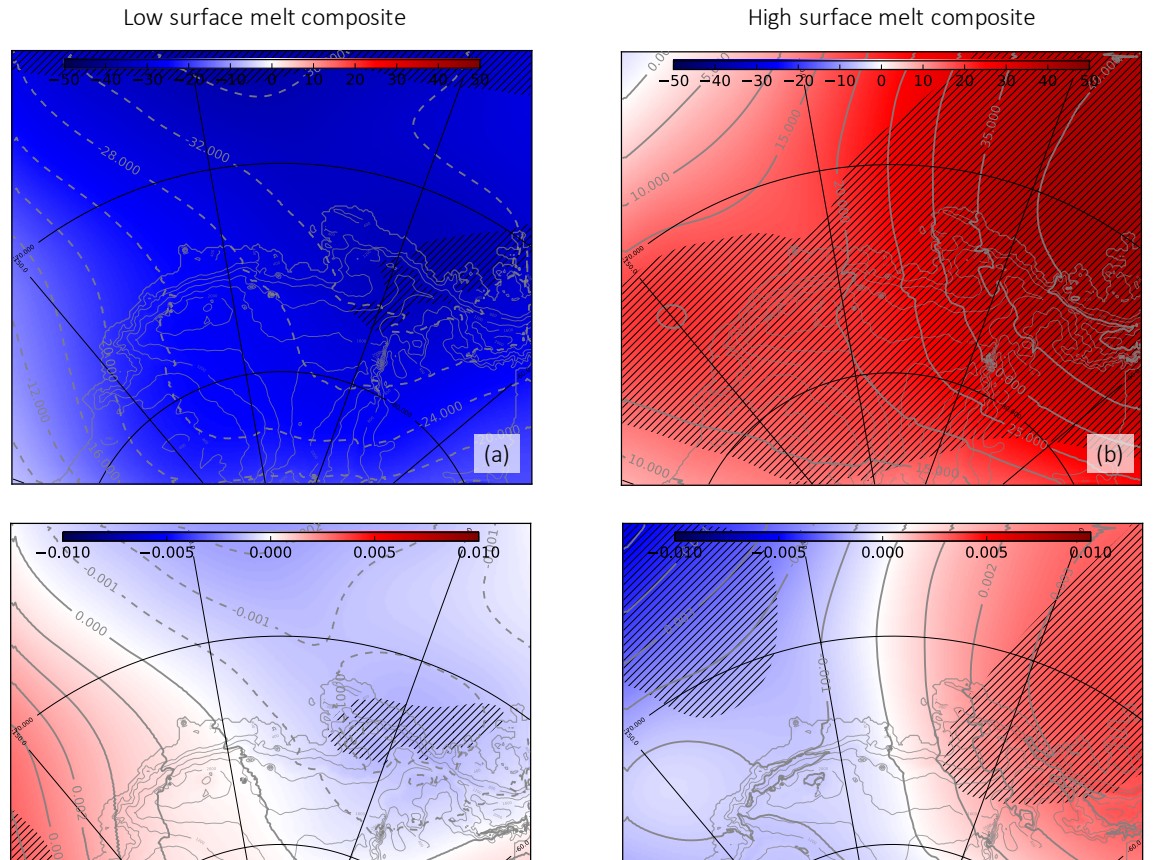

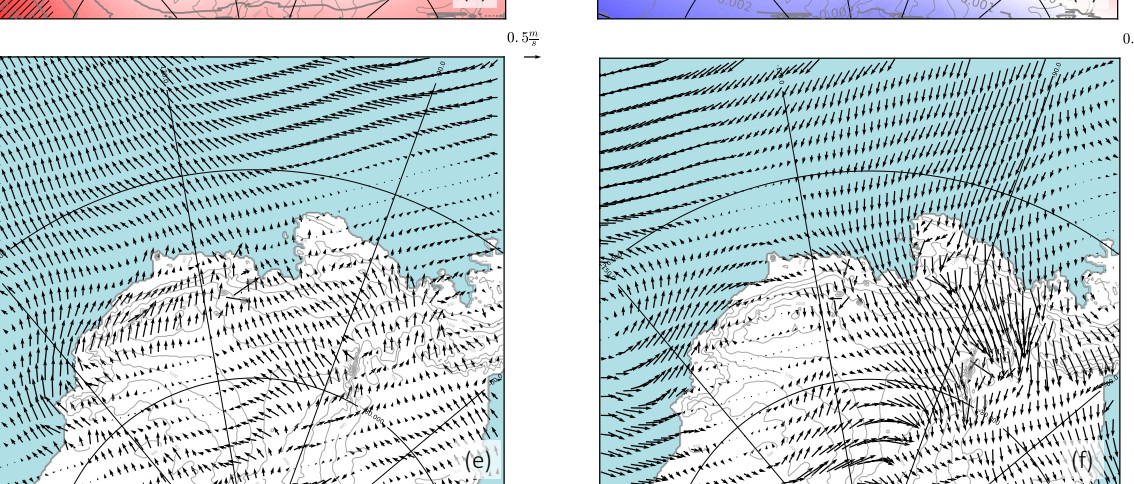

**Figure 12 : (a,b) ) 500 hPa geopotential height (m) and (c,d) 500 hPa geopotential height divided by the domain-averaged value for each season and (e,f) 10m wind (m s⁻¹) anomalies during low-melt summers (left) and high-melt summers (right), scales of arrow lengths are shown near the upper right corner of panels (e) and (f). Anomalies are calculated as high/low composites minus the climatology over 1979-2017. Hatched area represents significance>90% calculated with Welch's *t*-test.**


Now that we have described the mechanisms in play for summers with high and low SMB or surface melt rates, we investigate the connections between the leading modes of climate variability (ENSO, SAM and ASL variability) and summer SMB and surface melting over the individual Amundsen drainage basins (shown in Fig.1).


In line with the previous composite analysis for high and low SMB composites, the SMB in all the drainage basins is anti-correlated to the ASL longitudinal position (Table 3, 4[th] column). This anti-

correlation has little statistical significance for Abbot and Cosgrove, but for Dotson and Thwaites, the
ASL longitudinal position explains nearly 40% of the SMB interannual variance (explained variance
given by square correlations). The ENSO-SMB relationship has moderate levels of statistical significance,
with positive SMB correlations to -SOI for all basins, but a part of SMB variance explained by ENSO
that remains below 16% (Table 3, 2nd column). -SOI and the ASL longitudinal location are not
significantly connected together (Table 1), therefore their connection to SMB can be considered as
independent from each other. Finally, the SMB is significantly correlated to neither the ASL relative
central pressure (Table 3, 5th row) nor the SAM index (Table 3, 3rd column) for all the basins. To better
describe interplays, we also calculate a multi-linear regression of SMB on the 4 indices (last column of
Table 3). Accounting for several indices increases the explained SMB variance compared to a single
index, indicating an interplay of the ASL and ENSO. Overall, 16 to 49% of the summer SMB variance
(increasing westward) can be explained by a linear combination of the climate indices.

**Table 3 : Correlation R between ENSO, SAM, and ASL indices and the SMB over individual drainage basins in austral summer. The statistical significance (Welch's t-test) is written within brackets. The last column shows the correlation of a multi-linear regression to the 4 indices using a least absolute shrinkage and selection operator (LASSO, Tibshirani, 1996)**

| Drainage Basins | -SOI vs SMB | SAM index vs SMB | ASL longitudinal location vs SMB | ASL relative central pressure vs SMB | multi-linear regression |
|---|---|---|---|---|---|
| Abbot | 0.25 (87%) | 0.14 (59%) | -0.15 (65%) | -0.01 (3%) | 0.40 |
| Cosgrove | 0.26 (88%) | 0.16 (65%) | -0.21(80%) | 0.08 (36%) | 0.46 |
| Pine Island | 0.32 (95%) | 0.03 (17%) | -0.25 (87%) | -0.17 (69%) | 0.47 |
| Thwaites | 0.33 (96%) | 0.02 (8%) | -0.45 (99%) | -0.10 (47%) | 0.57 |
| Crosson | 0.40 (99%) | -0.00 (2%) | -0.53 (99%) | -0.14 (60%) | 0.66 |
| Dotson | 0.36 (97%) | 0.00 (2%) | -0.61 (99%) | 0.15 (65%) | 0.70 |
| Getz | 0.30 (93%) | -0.15 (62%) | -0.64 (99%) | 0.27 (90%) | 0.68 |

We now investigate similar relationships, but with surface melt rates instead of SMB. By contrast
to SMB, the surface melt connection to the ASL relative central pressure is stronger than its connection
to the ASL longitudinal position (Table 4, 4th and 5th columns), which again highlights the two distinct
mechanisms explaining high/low melt rates vs high/low SMB. The part of the melt rate variance explained
by the ASL relative central pressure increases westward, from 12% for Abbot to 21% for Getz. Even
though the effect of the ASL central pressure dominates, there is still a moderate anti-correlation between
melt rates and the ASL longitudinal position, suggesting that the mechanism explaining high/low SMB
can explain a small part of the melt rate variance (less than 10%). In a similar way as SMB, SOI explains

less than 9% of the melt rate's variance, with moderate statistical significance (Table 4, 2nd column), and as for summer SMB there is no significant relationship to the SAM. We have repeated the calculations considering the number of melt days, and we find very similar results in terms of correlations (Table 4, 2nd line in each row). Relatively similar conclusions can be drawn from observational estimates of number

of melt days (values in italic in Table 4), except that satellite estimates indicate a stronger correlation to -SOI, even exceeding the correlation to the ASL central pressure in the case for most drainage basins (the variance explained by -SOI reaching 25%). As the SAM index is significantly anti-correlated to ENSO (Table 1), the stronger melt-SOI correlation in the observational products goes together with a stronger melt-SAM anti-correlation than in our simulations. To better describe interplays, we also calculate a

multi-linear regression of melt rates on the 4 indices (last column of Table 4). Accounting for several indices increases the explained melt rate variance compared to a single index, which indicates an interplay of the fours modes of variability.  Overall, 21 to 30% of the summer melt rate variance can be explained by a linear combination of the climate indices.

        The part of explained variance never exceeds 50% of the summer melt and SMB variance.

Possible reasons for this are (i) the modes of variability do not explain all the variance locally; for example, the leading EOF of SST in the Equatorial Pacific (representing ENSO) only accounts for 50 to 70% of the SST variance (e.g. Roundy, 2014) meaning that the tropical convection thought to influence Antarctica is not completely described by SOI or NINO3.4; (ii) assuming that a large part of the tropospheric circulation variability is explained by ENSO, SAM and ASL indices, there are reasons why

the connection may be weaker for SMB and surface melting because of their non-linear dependence on sea ice and evaporation in coastal regions, the evolution of snow properties, etc; (iii) strong modulation of the southeast Pacific extratropical circulation by Rossby wave train is not only due to the existence of El Niño events but also depends on the exact spatial distribution of deep convection in the tropical central Pacific and to the strength of the polar jet (Harangozo, 2004) (iv) a part of the variability of SMB and

melting may be stochastic, i.e. not necessarily driven by variability with spatio-temporal coherence at large scales.


**Table 4 : Correlation R between -SOI, SAM, and ASL indices and MAR surface melt rates (bold), MAR number of melt days (regular), number of melt days from satellite products (italic, first value for Nicolas et al. (2017) and second for Picard et al. (2007),over individual ice-shelves in summer. The statistical significance (Welch's t-test) is written within brackets. The last column shows the correlation of a multi-linear regression to the 4 indices using a least absolute shrinkage and selection operator (LASSO, Tibshirani 1996).**


| Drainage Basins | -SOI | SAM index | ASL longitudinal location | ASL relative central pressure | multi-linear regression |
|---|---|---|---|---|---|
| Abbot | **0.23 (84%)** | **-0.05 (24%)** | **-0.25 (86%)** | **0.35 (97%)** | **0.46** |
|  | 0.25 (86%) | -0.04 (19%) | -0.23 (84%) | 0.30 (93%) | 0.44 |
|  | *0.37 (97%)* | *-0.22 (79%)* | *-0.29 (91%)* | *0.32 (94%)* | *0.49* |
|  | *0.37 (98%)* | *-0.18 (71%)* | *-0.18 (72%)* | *-0.24 (92%)* | *0.47* |
| Cosgrove | **0.24 (86%)** | **-0.08 (36%)** | **-0.30 (93%)** | **0.37 (98%)** | **0.50** |
|  | 0.25 (87%) | -0.06 (29%) | -0.29 (92%) | 0.32 (95%) | 0.47 |
|  | *0.37 (97%)* | *-0.20 (76%)* | *-0.37 (97%)* | *0.32 (94%)* | *0.52* |
|  | *0.38 (98%)* | *-0.25 (87%)* | *-0.16 (65%)* | *0.27 (90%)* | *0.46* |
| Pine Island | **0.30 (86%)** | **-0.07 (33%)** | **-0.31 (94%)** | **0.38 (98%)** | **0.54** |
|  | 0.29 (92%) | -0.03 (13%) | -0.34 (96%) | 0.35 (97%) | 0.55 |
|  | *0.48 (99%)* | *-0.29 (91%)* | *-0.21 (78%)* | *0.42 (99%)* | *0.62* |
|  | *0.44 (99%)* | *-0.19 (75%)* | *-0.13 (56%)* | *0.37 (98%)* | *0.59* |
| Thwaites | **0.29 (92%)** | **-0.13 (56%)** | **-0.25 (87%)** | **0.39 (98%)** | **0.51** |
|  | 0.35 (95%) | -0.11 (43%) | -0.19 (69%) | 0.51 (99%) | 0.67 |
|  | *0.48 (99%)* | *-0.23 (81%)* | *-0.11 (45%)* | *0.29 (91%)* | *0.55* |
|  | *0.44 (99%)* | *-0.28 (89%)* | *-0.06 (26%)* | *0.26 (87%)* | *0.52* |
| Crosson | **0.28 (91%)** | **-0.14 (60%)** | **-0.23 (84%)** | **0.41 (99%)** | **0.51** |
|  | 0.29 (86%) | -0.08 (30%) | -0.11 (42%) | 0.40 (97%) | 0.52 |
|  | *0.48 (99%)* | *-0.35 (95%)* | *-0.20 (76%)* | *0.39(98%)* | *0.61* |
|  | *0.35 (96%)* | *-0.35 (96%)* | *-0.10 (45%)* | *0.41 (98%)* | *0.52* |
| Dotson | **0.27 (90%)** | **-0.14 (60%)** | **-0.24 (86%)** | **0.42 (99%)** | **0.52** |
|  | 0.26 (86%) | -0.13 (54%) | -0.25 (86%) | 0.44 (99%) | 0.53 |
|  | *0.36 (95%)* | *-0.27 (84%)* | *-0.03 (11%)* | *0.36 (94%)* | *0.52* |
|  | *0.33 (93%)* | *-0.28 (86%)* | *0.13 (51%)* | *0.32 (91%)* | *0.50* |
| Getz | **0.22 (82%)** | **-0.16 (65%)** | **-0.26 (88%)** | **0.46 (99%)** | **0.53** |
|  | 0.22 (82%) | -0.16 (67%) | -0.29 (92%) | 0.46 (99%) | 0.54 |
|  | *0.50 (99%)* | *-0.42 (99%)* | *-0.24 (84%)* | *0.41 (99%)* | *0.64* |
|  | *0.34 (96%)* | *-0.41 (98%)* | *-0.15 (63%)* | *0.34 (96%)* | *0.46* |

## 4 Discussion

The composite analysis and the correlation of SMB and melt rates to the ASL indices gives a consistent picture. Summers tend to be associated with high SMB when the ASL migrates westward and southward because this places the northerly flow (ASL eastern flank) over the Amundsen Sea, thereby increasing the southward humidity transport and snowfall. This corresponds to the large-scale features described by Hosking et al. (2013) but is here described for the SMB of individual drainage basins. By contrast,

longitudinal migrations of the ASL are not the main driver of surface melting variability, as previously noted by Deb et al. (2018). Summers tend to be associated with high surface melt rates when the Amundsen/Bellingshausen region experiences blocking, i.e. anticyclonic conditions, which tends to decrease the climatological southerly flow (western flank of the ASL), and to favor marine air intrusions that make cloud cover denser with increase downward longwave radiation, as described by Scott et al.

(2019).

While the role of the ASL now appears to be quite clear, the exact impact of ENSO on SMB and surface melt rates remains elusive. Earlier studies analyzing the impact of ENSO on precipitation in West Antarctica had difficulties understanding the mechanisms and the robustness of the signal, because they had to rely on relatively short observation and reanalysis periods (Bromwich et al., 2000; Cullather et al.,

1996; Genthon and Cosme, 2003). Using a dedicated SMB model over a longer time period, we have shown here that the ENSO-SMB relationship in austral summer exists, but it is relatively weak as SOI alone cannot explain more than 16% of the interannual variance in summer SMB. The relationship between ENSO and the number of melt days was identified by Deb et al. (2018) using both regional simulations and a satellite product. It was then thoroughly described by Scott et al. (2019) who found that

SOI could explain 20% of the melt variance when considering all the Amundsen ice shelves together and using satellite products (correlation of 0.45 in their Table 3). While we obtain similar results as Scott et al. (2019) when using the number of melt days derived from satellite products, both the number of melt days and the melt rates simulated by MAR indicate less variance explained by SOI, that is, between 5% and 9% for the individual drainage basins. Our MAR simulations certainly contain biases in the

representation of the melting process and the way it affects surface properties such as albedo and roughness, but it is also possible that the number of melt days derived from microwave satellite data is biased due to variability in surface conditions, percolation within fresh snow, meltwater ponding (observed on Pine Island, Kingslake et al., 2017), and satellite overpass time (Tedesco, 2009 ; Scott et al., 2019). More work will be needed to understand these differences.

Numerous publications have explained the remote effects of ENSO on the West Antarctic climate through Rossby wave trains that connect the convective anomalies associated with ENSO in the equatorial Pacific to Antarctica (e.g., Yuan and Martinson, 2001). However, austral winter and spring conditions are more favorable for Rossby wave trains to be formed and to propagate to high southern latitudes than summer conditions (Harangozo, 2004; Lachlan-Cope and Connolley, 2006; Ding et al., 2011 and

references therein). The poleward propagation of tropically sourced Rossby waves in summer is indeed inhibited by the strong polar front jet in the South Pacific sector at that time of the year, which leads to Rossby wave reflection away from the Amundsen Sea region (Scott Yiu and Maycock, 2019). This lack of direct connection in summer was supported by Steig et al. (2012) who found weakest correlations between NINO3.4 and wind stress anomalies in the Amundsen Sea in DJF compared to other seasons. Therefore, we investigated possible lags in the relationships to ENSO. While ENSO peaks in DJF, it starts to develop in MAM, as indicated by the growing SOI auto-correlation from 9 to 6 months lag (Fig. 13a). The first implication of this is that any signal correlated to SOI in DJF will be correlated to SOI in the previous JJA without the need for a lagged physical mechanism. Nevertheless, the correlation between SMB or melt rates in DJF and SOI in the preceding JJA is higher than the synchronous correlation for all the drainage basins (solid curves in Fig. 13b-h), which suggests that the lagged relationship is not only a simple statistical artifact. The results of Ding et al. (2011) and Steig et al. (2012) suggest that there could be a lagged mechanism whereby ENSO would influence West Antarctica in austral spring or winter, with a delayed response of SMB and melting in the following austral summer. The number of melt days derived from satellite data also gives 6-month lagged correlations to SOI that are as high or higher than synchronous correlations for most ice shelves (dashed curves in Fig. 13b-h).

We now discuss possible explanations for this lag. As mentioned previously, the Rossby wave trains connecting the Equatorial Pacific to Antarctica are expected to develop within a few weeks in response to ENSO convective anomalies (e.g. Hoskins and Karoly, 1981; Mo and Higgins, 1998; Peters and Vargin, 2015). Therefore, the lag has to come from anomalies stored in a slower medium, such as snowpack, ocean, or sea ice. Snow surface melting in DJF is neither correlated to the temperature of snow layers within the first 2m in the previous months (not shown), nor to the snow accumulated over the previous months (not shown). This indicates that heat diffusion in snow or preconditioned porosity or albedo of snow are not responsible for the 6-month lag. By contrast, we find that El Niño events in JJA significantly reduce the sea ice cover in the following DJF (Fig. 14). This is reminiscent of Clem et al., (2017) who found stronger lagged correlation between SON ENSO and DJF sea ice cover than synchronous correlation in DJF, with consequences on summer air temperatures. We suggest two possible explanations for this lagged ENSO-sea-ice relationship. First, it could be slowly advected from the Ross Sea. Pope et al., (2017) indeed found that El Niño events developing in MAM created a dipole of sea ice anomalies, with decreased (increased) concentration in the Ross Sea (Amundsen and Bellingshausen Seas). Using a novel sea ice budget analysis, they showed that the decreased concentration in the Ross Sea was then advected eastward, reaching the Amundsen Sea in SON and DJF.

There is also another possible pathway for lagged ENSO/sea-ice relationship. The zonal wind stress over the Amundsen Sea continental shelf break is a good proxy for the transport of Circumpolar Deep Water (CDW) onto the continental shelf (Thoma et al., 2008; Holland et al., 2019). Steig et al., (2012) noted significant correlations between that wind stress and ENSO in JJA and SON but not in DJF. All

these studies as well as Paolo et al., (2018) pointed out scales of a few months for the buildup and advection of CDW on the continental shelf then into the ice shelf cavities where they produce basal melting, and Paolo et al., (2018) reported correlations between ENSO and ice shelf thinning 6 months later. As stronger ice-shelf melt rates tend to decrease sea ice in this region due to the entrainment of warm CDW towards the surface (Jourdain et al., 2017; Merino et al., 2018), the connection through CDW intrusions may also explain a part of the lag between ENSO and DJF sea ice in the Amundsen Sea. We suggest that both mechanisms (eastward advection of sea ice anomalies and anomalous intrusions of CDW) may explain the 6-month lag between DJF SMB or melting and ENSO, and we leave the details of the ocean/sea-ice processes for future research.

Beyond the ASL and ENSO, we also find that the SAM is not significantly related to summer SMB and surface melt over individual drainage basins at interannual time scales, which agrees with Deb et al. (2018). This may appear contradictory to the results obtained by Medley and Thomas (2019), showing that the positive SAM trend from 1957 to 2000 largely explains the pattern of annual SMB trends over the Antarctic ice sheet. First of all, their residual SMB trend (i.e. not related to SAM) is particularly strong in the Amundsen Sea Embayment (their Fig. 1e), highlighting that only a part of the SMB trend in that region may be related to the SAM trend. The multi-decadal SAM trend is also related to ozone depletion and emissions of greenhouse gases, and the interannual SAM variability may have different characteristics and impacts on SMB. Furthermore, the absence of SMB-SAM relationship in our MAR simulations is specific to the austral summer, which represents 15% of the annual SMB, and correlations are more significant for the other seasons (Table S3 in supplementary material). Therefore, the significant SAM-SMB relationship suggested by Medley and Thomas (2019) for annual SMB are not necessarily in contradictory to our results. Lastly, previous studies have suggested that the SAM-ENSO anti-correlation may diminish the impact of ENSO on surface melting and SMB. Partial correlations used to disentangle the SAM and ENSO influences on SMB do indicate a slightly stronger SMB-ENSO correlation when the effect of SAM is removed (in particular for Abbot and Cosgrove, see 2[nd] and 3[rd] columns of Table 5), but the effect is relatively small. For melt rates, the SAM modulation is very weak for all the basins (Table 5, 4[th] and 5[th] columns).

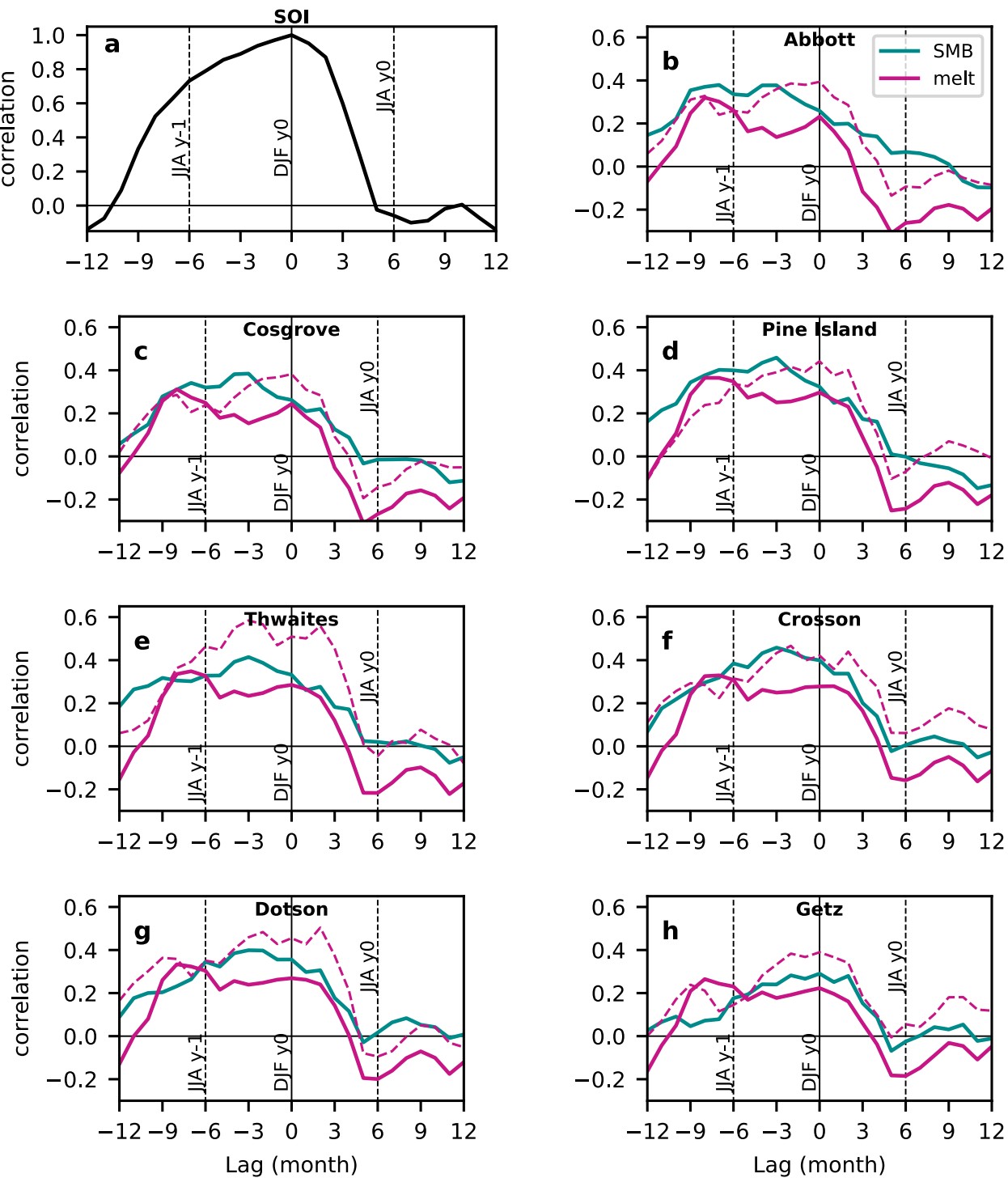

**Figure 13 : Correlation between lagged 3-month averaged -SOI (i.e. DJF at zero lag, previous JJA at -6 lag) and (a) DJF SOI, (b-h) simulated SMB and melt rates in individual drainage basins. The dashed curves correspond to the number of melt days derived from satellite data by Picard et al. (2007).**

**Table 5 : Partial Correlation of -SOI vs SMB or melt rates, removing the influence of SAM (columns 2 and 4). Corresponding full correlations are indicated in columns 3 and 5 (same as Table 3 and Table 4).**

| Drainage Basins | Partial correlation -SOI vs SMB (without SAM) | -SOI vs SMB | -SOI vs SMB | Partial correlation -SOI vs surface melt (without SAM) | Correlation -SOI vs surface melt |
|---|---|---|---|---|---|
| Abbot | 0.36 | 0.25 | 0.21 | 0.23 | 0.23 |
| Cosgrove | 0.37 | 0.26 | 0.21 | 0.23 | 0.24 |
| Pine Island | 0.38 | 0.32 | 0.26 | 0.30 | 0.30 |
| Thwaites | 0.38 | 0.33 | 0.23 | 0.25 | 0.29 |
| Crosson | 0.45 | 0.40 | 0.29 | 0.24 | 0.28 |
| Dotson | 0.40 | 0.36 | 0.25 | 0.23 | 0.27 |
| Getz | 0.26 | 0.30 | 0.18 | 0.17 | 0.22 |

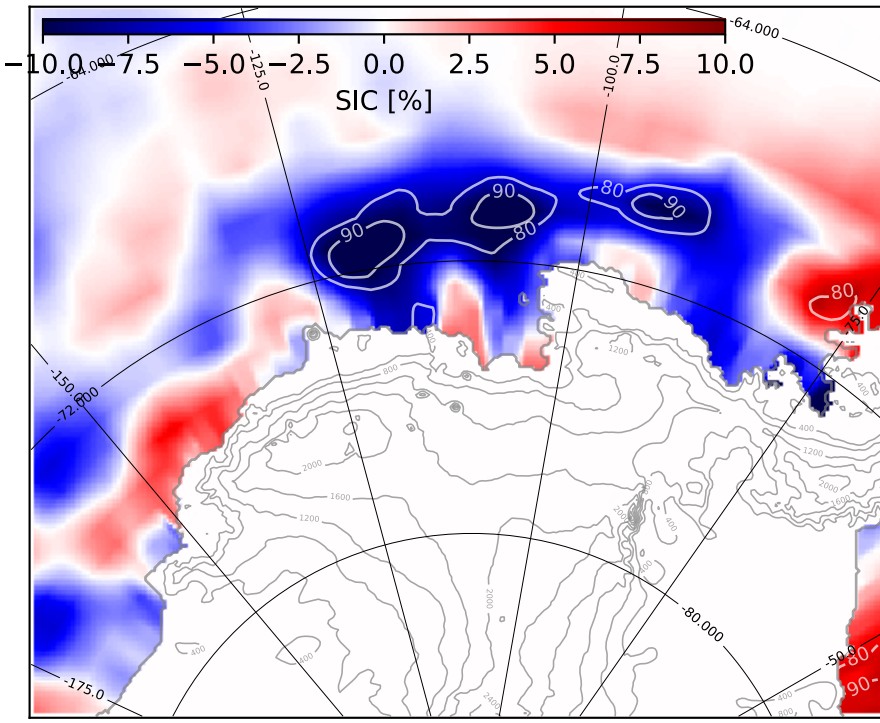

**Figure 14 : Summer sea ice cover (%) anomaly (composites minus the climatology over 1979-2017) during El Niño events in JJA (6 month before). Contours represent significance with Welch's _t_-test.**

Lastly, we discuss the relationship between surface melt and snowfall over the ice shelves of the Amundsen sector (last rows of Tab. 2). According to Table 2, runoff is null over all the ice shelves, which means that the firn is never saturated. In other words, all surface meltwater and rainfall refreeze within the firn. This is consistent with Pfeffer et al. (1991) who estimated that the melt rate needed to saturate the firn with water and lead to hydrofracturing can be estimated as 0.7 times the snowfall rate (both melt and snowfall rates expressed in $kg.m^{-2}.s^{-1}$ or mm.w.e). This indicates that meltwater ponding and complex surface hydrological flows are unlikely to develop over West Antarctic ice shelves under the current climate. To reach saturation at the scale of the entire ice shelf in the future (and therefore to initiate hydrofracturing), the 0.7 ratio of Pfeffer et al. (1991) suggests that melt rates would need to be multiplied by 2.5 (Cosgrove) to 40 (Crosson) compared to present conditions.

## 5 Conclusion

In this paper we have analyzed possible drivers for summer surface melt and SMB interannual variability over the last decades in the Amundsen sector, West Antarctica. For this, we have simulated the 1979 to 2017 period with the regional atmospheric model, MAR. We have first evaluated our model configuration in comparison to observational products (i.e. AWS, airborne-radar and firn-core SMB, melt days from satellite microwave, and melt rates from satellite scatterometer). MAR gives good results for near-surface temperatures (mean overestimation of 0.10°C), near-surface wind speeds (mean underestimation of 0.42 m s$^{-1}$), and SMB (local relative bias < 20% over the Thwaites basin). The mean surface melt rate over the Amundsen Sea region is underestimated by 18% compared to the estimates derived from QuickSCAT (Trusel et al., 2013), and the interannual variability of surface melting is relatively well reproduced in terms of melt rate (R=0.80) or number of melt days (R=0.43 to 0.69 depending on the satellite product) as also found by previous study using the same MAR version (i.e Datta et al., 2019). Similar underestimation was also found in another regional atmospheric model of the Amundsen region (underestimation of 30–50% found by Lenaerts et al., 2017). Overall, our results indicate that MAR is a suitable tool to study interannual variability in the Amundsen sector.

Then, we have analyzed the interannual variability of summer SMB. The strongest summer SMB occurs over Thwaites and Pine Island glaciers when the ASL migrates far westward (by typically 30°) and southward (by typically 3-4°). This promotes a southward flow on the Eastern flank of the ASL, towards the glaciers, with resulting increased moisture convergence, precipitation, and therefore SMB. Our study hence provides further support for the connection between Antarctic precipitation and the ASL longitudinal position that was previously described by Hosking et al. (2013) based on the ERA-interim reanalysis. In terms of climate indices, this corresponds to an anti-correlation between SMB and the ASL longitudinal position. This anti-correlation is found for all the drainage basins of the Amundsen Sea Embayment, and the part of the SMB variance explained by the ASL longitudinal migrations ranges from

2 % to 41 % (increasing westward). A small part of the SMB variance is also related to ENSO, with higher SMB during El Niño events and lower SMB during La Niña, but less than 8% of the SMB variance is explained by ENSO variability. This SMB connection to ENSO is independent from its connection the ASL longitudinal position.

We have also analyzed the interannual variability of summer surface melt rates. Strongest surface melting occurs over Thwaites and Pine Island glaciers when the ASL undergoes an anticyclonic anomaly (likely the signature of blocking activity), which is visible through anomalies of the ASL relative central pressure. Such an anomaly promotes a southward anomaly of near-surface winds anomaly and moisture convergence over the Amundsen Sea Embayment. As recently described by Scott et al. (2019), this leads

to increased cloud cover and downward longwave radiation, which in turns increases surface melting. As for SMB, we do not find that surface melt rate variability in our simulations is strongly connected to ENSO as it does not explain more than 9% of the total variance in simulated summer surface melt rate (or 12% of the number of melt days). By contrast and for unknown reasons, the variance in number of melt days derived from satellite products indicates that as much as 25% of the variance in these products

could be explained by -SOI.

We also suggest that at least a part of the ENSO-SMB and ENSO-melt relationships in summer is inherited from the previous austral winter (JJA). Rossby wave trains generated by convective anomalies related to developing El Niño events in austral winter significantly affect the Antarctic region and we suggest that this has some impact on SMB and surface melting in the Amundsen sector 6 months later.

Such delay could either be related to sea ice anomalies generated by ENSO in the Ross Sea in austral winter and taking 6 months to be advected to the Amundsen Sea (Pope et al., 2017), or to marine intrusions of Circumpolar Deep Water that are favored by El Niño events in austral winter (Steig et al., 2012). Circumpolar Deep Water may take 6 months to reach ice shelf cavities where increased basal melting favors the entrainment of this water towards the ocean surface (Jourdain et al., 2017). It should

nonetheless be noted that even accounting for this 6-month lag, the influence of ENSO on summer SMB and melt rates remains weak, not explaining more than 15% variance.

Lastly, we propose that the rate of surface water needed to saturate the firn and lead to hydrofracturing has to increase by a factor of 2.5 to 40 depending on the ice shelf. Such increase could be reached under strong warming scenarios given the exponential temperature dependence described by Trusel et al.,

(2015) , although snowfall is also expected to increase (Krinner et al., 2008; Agosta et al., 2013; Ligtenberg et al., 2013; Lenaerts et al., 2016; Palerme et al., 2017), requiring even more meltwater to reach saturation. In their projections, Kuipers Munneke et al. (2014) found that the Western part of Abbot as well as Cosgrove could become water-saturated before the end of the 22$^{nd}$ century, but the other ice shelves of the Amundsen sector remained non-saturated.    Further work will be needed to assess the

robustness of these projections, with other firn models and global projections.

**Code and data availability:** The MAR code (version 3.9.1) is available on the MAR website (http://mar.cnrs.fr/), outputs from the Amundsen simulation presented in this study are available on https://doi.org/10.5281/zenodo.2815907.


**Author contributions:** The study was designed by Marion Donat-Magnin and Nicolas C. Jourdain. Set-up of the MAR domain configuration was made by Marion Donat-Magnin, Cécile Agosta, Amine Drira and Nicolas C. Jourdain. Cécile Agosta, Xavier Fettweis, Hubert Gallée, Christoph Kittel and Charles Amory developed and tuned the MAR model for Antarctica, and they contributed to improving and
interpreting our simulations. Christoph Kittel developed the scripts used to compare MAR to AWS data. Jonathan D. Wille and Vincent Favier contributed to the interpretation of our results related to interannual variability. All the authors significantly contributed to this manuscript.

**Competing interests:** The authors declare that they have no conflict of interests.


**Acknowledgements:** This work was funded by the French National Research Agency (ANR) through the TROIS-AS (ANR-15-CE01-0005-01) project. The development of MAR was partly funded by Labex OSUG@2020 (ANR10 LABX56) through the "Tout le Monde se MAR" project. All the computations presented in this paper were performed using the GRICAD infrastructure (https://gricad.univ-grenoble-
alpes.fr), which is partly supported by the Equip@Meso project (ANR-10-EQPX-29-01) of the program "Investissements d'Avenir" supervised by ANR, by the Rhône-Alpes region (GRANT CPER07_13 CIRA: http://www.ci-ra.org) and by France-Grilles (http://www.france-grilles.fr). We thank G. Picard, L.Trusel, J. Nicolas, Y. Wang and B.Medley for making their data available.

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
