# Peer review of "Interannual Variability of Summer Surface Mass Balance and Surface Melting in the Amundsen Sector, West Antarctica"

_The Cryosphere, 2019_

## Referee Comment (RC1) · Anonymous Referee #1 · 28 Jun 2019

Summary – The authors present a significant amount of work on MAR model validation and ultimate evaluation of drives of summer surface mass balance (SMB) and melt over the Amundsen Sea sector of Antarctica. They provide ample background of previous studies of the climate in West Antarctica, MAR model description as well as numerous observations used in the evaluation, full descriptions of the climate indices investigated, and, of course, their findings. The model reproduced temperature, wind speed, SMB, and melt intensity and days, and thus, the authors indicate that MAR is sufficient to evaluate drivers of SMB and melt in this sector. Specifically, they find that the longitudinal position of the Amundsen Sea Low (ASL) is the primary driver (relative to ENSO and the Southern Annular Mode, SAM) of SMB variability, whereas the

variability in the ASL central pressure and to a lesser extent ENSO drive melt with increasing control moving westward. The SAM was not strongly related to either SMB or melt. The authors finally surmise that there might be a 6-month lag between ENSO and West Antarctic climate via either sea ice anomalies and their transport or a lag in Circumpolar Deep Water intrusions.

The paper is generally well written and very thorough (much appreciated!) and is organized in a logical fashion. Many of the insights are not necessarily new; however, the paper presents new model work and output and attempts to present a cohesive picture of drivers of West Antarctic climate. All of the analyses presented appeared appropriate and well thought out, and the tools used were appropriate. The work presented is very thorough yet easy to understand, making it a pleasure to read.

Major Comments

One important consideration that is missing is a description of why summer SMB is critical and how it relates to the annual SMB. December-January-February (DJF) clearly are the relevant months for surface melting, but I think there should be more discussion as to why the paper specific isolated summer SMB. Specifically, melt is the highest in DJF, but snowfall is typically the lowest in DJF (Lenaerts et al., 2012). Please consider evaluation of winter (or other seasons of SMB relevance) or add language justifying the importance of summer SMB.

The relationship between melt and SMB is not investigated. The paper provides background on the importance of the role of melt on hydrofracture of ice shelves and potential rapid disintegration of an ice shelf, but it does not discuss the role of firn pore space. According to Table 2, nearly all of the surface melt refreezes within the firn column, so this mechanism should be introduced as well. The paper also notes that into the future there will be more snowfall and melt, but did not mention that the enhanced snowfall could potentially also provide more pore space for meltwater infiltration and refreezing. Please consider additional discussion of the role of SMB (or snowfall) on

providing addition pore space for surface melt.

There is no discussion on the relatively small proportions of variance explained by the climate indices. For instance, over western WAIS 20-40% of the summer SMB variability can be explained by the ASL longitude; however, it explains <6% of the Abbot, Cosgrove, and Pine Island catchments. None of the indices is significantly correlated with SMB over those catchments. Thus, the impact of ASL longitude is only relevant from Thwaites moving westward. The paper should make this clear and also potentially investigate other drivers of change for the eastern catchments or at least add clarifying statements that the drivers in eastern WAIS are unknown and potentially postulate why. Along similar lines, while ASL central pressure is a clear control on all catchments, the explained variance range from 12-21%, suggesting that there are additional factors at play when it comes to surface melt. Would investigation of multiple regression with the different indices help clarify how they interplay (for example, perhaps the combination of some movement and strengthening or weakening of the ASL is more strongly related). Please consider adding more multivariate relationships and discuss other potential influences on meltwater production since only a small portion is explained.

The postulation of potential lags is not adequately investigated. The hypothesis regarding sea ice reduction and transport from the Ross Sea could be tested as MAR using the sea-ice concentration from ERA-Interim. Thus, please consider adding analysis of sea ice concentrations to support this postulation. Although not as clear cut, intrusion of marine CDW could be evaluated by looking at the effective wind stresses as done by Steig et al. just off the continental shelf and attempt to quantify a six-month lag between strong wind events and surface melt. Also, there is no mention of potential preconditioning of the snowpack/firn for melt. An additional important variable in control of surface melt in the summer is the amount of snow that fell the prior winter, and it should be added to the analysis presented and included in Table 4. This signal might not matter at all, but also could lead to misinterpretation of an ENSO lag. Please consider all potential snowpack preconditioning variables that might explain melt from

year-to-year.

Minor Comments

Line 16 – change to "Amundsen Sea glaciers"

Line 58 – change "underlaying" to "underlying"

Line 114 – remove the ';' at the beginning of the line

Line 140 – change "estimates" to "estimate"

Line 185-186 – remove the sentence "These data were collected over the Thwaites and Pine Island basins." as it is redundant with Lines 181-182.

Section 2.3: Are these indices derived from ERA-Interim for consistency with the MAR output? If not, please state that and justify their use.

Line 273 – Are "overestimate" and "underestimate" confused? Shouldn't it be "The model tends to underestimate and overestimate highest and lowest wind speeds"?

Line 299 – Remove "(Medley et al. 2013, 2014)" as it is already mentioned in the sentence.

Line 382 - add "is" after "mechanism"

Figure 10/11 – Please add in the legend that blue represents moisture convergence for clarity.

Paragraph beginning with 530 – Perhaps it is important to mention here that DJF makes up the smallest percentage of annual accumulation, so it is not surprising that the findings do not match Medley and Thomas.

---

## Short Comment (SC1) · 16 Jul 2019

This is a comprehensive analysis.

One major shortcoming is that it really underplays the comparisons of the present results with those obtained by Deb et al. (2018) also based on regional climate modeling for 1979-2015 summers where a leading conclusion is:

"El Niño episodes during austral summer drive warmer conditions over Amundsen Sea Embayment ice shelves that cause enhanced surface melting".

El Niño influences play a relatively minor role in the current analysis. The explanation

likely lies in the discussion on lines 475-486.

I didn't think the analysis for a lagged relation between El Niño forcing SMB/melting (Fig. 13) to be very compelling, at best possible.

I don't understand what is meant by humidity divergence (Figs. 10 and 11). Normally one evaluates moisture transport divergence in relation to P-E. Please clarify.

---

## Referee Comment (RC2) · Anonymous Referee #2 · 17 Jul 2019

This paper presents results from the regional climate model MAR run for the Amundsen Sector of West Antarctica. The paper is well written and thorough, although some parts the paper needs improvement.

After reading the paper and collecting my points of concerns, I've read the other review and the comment of David Bromwich. I agree with their major concerns and these concerns have to be addressed. Additionally, I have the following major comments:

It needs to be addressed why SMB summer is discussed and not the annual SMB. I can imagine a reason, but this - or any other - reason is not given.

Although the patterns in Figures 8-12 are logic and reasonable, its worrisome that most

of the signals showed are insignificant. Try to get a better understanding of the significance. For example, for geopotential fields the gradient matters more than the value, so you might take a "relative elevation" approach similar to the ASL central pressure. You might also try a different method to determine significance, for example, bootstrapping. If the patterns remain mostly insignificant it implies that the shown patterns do occur during high/low melt/SMB but not necessarily lead to high/low melt/SMB.

I'm not convinced that humidity convergence @ 850 hPa is the best parameter to show. For SMB anomalies: As the moisture holding capacity of air is not that big, the convergence is directly linked to precipitation generation. Added compared to SMB is a whole bunch of noise due to variations in the elevation of the 850 hPa level and noise is added by apparently near stationary numerical waves. I would be more interested to see anomalies in the temperature @ 700 hPa / 850 hPa and vertical integrated moisture content fields. For melt anomalies: it likely boils down to that high melt years have also higher summer SMB although this relation might not be significant. Furthermore, the authors do show that cloudiness increases, but fail to prove that his is the only cause. To which extend is the higher melt due to cloudiness and which extend due to advection of warmer air? What is the anomaly of temperatures at 700 hPa? This anomaly can be easily included in Figures 12 a,b. I know temperature and cloudiness anomalies are likely covarying, so disentangling might be complicated. Helpful might be the MSSA technique (Plaut and Vautard, 1994; Allen and Robertson, 1996).

(line 527): CDW intrusions cannot be proven directly with the data from this manuscript (although SSTs and wind stress are available), but sea ice anomalies are available. It takes only a few steps to verify if the hypotheses are confirmed by data, so take those steps. And if the data does not confirm this hypothesis, that must be stated as well.

Minor comments

158: The sentence on the boundary relaxation is ambiguous: It could also mean that every 6 hours the state in at the boundaries "is forced back" to ERA-Interim values.

However, I presume that every time step fields are relaxed to ERA-Interim fields with 6-hourly temporal resolution. Rephrase to remove this ambiguity. Furthermore, add the boundary relaxation zone to the graph by using shading or something else and explain in the text how wide this zone was. From eg Fig 10 I conclude it was rather narrow, explain why or add a reference.

131: polar-oriented. Did you mean "polar adapted"? Oriented is not wrong but uncommon in this meaning.

224: I would prefer if these webpage-links could be included as references so that the text becomes less disturbed. But that's up to Copernicus to solve/decide on.

298: How this performance compares to other studies, thus MAR-full Antarctica and various RACMO2 products? Add a comment on this in the text.

321: It might be interesting to make a scatter plot of the modelled and interpolated QuickScat melt for their overlapping time period. You could color code the dots per ice shelve or drainage basin and even add regression lines per drainage basin. You don't discuss the few spots in West Antarctica where MAR gives high melt rates – do this. And have a look at https://www.the-cryosphere.net/13/1473/2019/tc-13-1473-2019.pdf if this might be a possible explanation for your model deviations too.

359: It would be nice if these high/low SMB/melt years as listed, maybe by adding symbols in figure 7.

363: In Figure 8 your plotting two differences per frame – that makes it harder to include signs of significance. Are these differences significant? Make a comment in the text and, if possible, find a way to display if deemed relevant.

378: Cloud cover could be a poorly performing parameter – I know models in which this is the case. Verify if you find similar/equivalent patterns in the vertical integrated cloud content (please add these figures in the rebuttal letter) and state in the manuscript if similar / equal patterns are found in the vertical integrated cloud content.

379: As snowfall exceeds the SMB due to sublimation, the "95%" in the quote is a bit odd. Rephrase.

425-427: This is not necessarily true. If positive SMB anomalies occur only if NINO34 is positive and ASL-longitude is negative, then their impact on SMB is not unrelated even though NINO34 and ASL-longitude are unrelated themselves. Thus, statistically verify your claim.

438: Would it not be more straightforward to see if there is a correlation between SMB and melt rates? And if not, state this.

Table S1: Add the numbers used in Fig 1 to the table – Yes, I know they are ordered from 1 to 41, but adding the number makes it just a slightly bit easier.

Fig 1: Consider to include excluded AWS stations in the figure using a different color, as long as they are on the map. Names are not needed.

Fig 2: The lines are not explained in the figure caption. Are the lines derived using normal fitting or perpendicular fitting techniques? Colors are not different enough to identify stations in the graph, so either use more distinguishable colors or simply don't try: give all lines the same color.

A drawback of a dot-plot is that you can't see differences in density once the dots form a continuous cloud. It might be worth the work to calculate the dot-density per (e.g.) 0.01 C-squared (Fig 2a) and plot this point density as contour graph on top of the dots. This added information on the point-density would make a statement like line 273-274 visible from the graph, the overestimation for low wind speeds is not well visible in the point cloud.

Fig 3: I'm not fond of the graphical solution to plot SMB in greyscale – details are hardly visible nor quantifiable. For example, I have no clue what the magnitude of the SMB from MAR is near the Medley data. Replace the grey by colors and add the basin delineation in a different manner. In all solutions, more detail must become visible for

SMB ranging from 200 to 500 mm w.e. per year.

Fig 5: Replace the grey by clear colors and extend the scale to higher values than 100 mm w.e. per year – this should be obvious as you do discuss these high melt values in the main text.

Fig 9: Contours in b are labelled with 0-2-5-8 intervals, but their regular spacing looks like 0-2.5-5-7.5. Check this. Hatching is not explained – should be done here too. Hatching line thickness varies with viewer.

Plaut, G., and R. Vautard, 1994: Spells of low-frequency oscillations and weather regimes in the Northern Hemisphere. J. Atmos. Sci., 51, 210–236, doi:https://doi.org/10.1175 Allen, M. R., and A. W. Robertson, 1996: Distinguishing modulated oscillations from coloured noise in multivariate datasets. Climate Dyn., 12, 775–784, doi:https://doi.org/10.1007

---

## Author Comment (AC1) · 27 Sep 2019

We thank Reviewer #1 for this constructive and motivating feedback. We agree with most of the following objections and have considered them in the revised manuscript.

Major Comments : C1 One important consideration that is missing is a description of why summer SMB is critical and how it relates to the annual SMB. December-January-February (DJF) clearly are the relevant months for surface melting, but I think there should be more discussion as to why the paper specific isolated summer SMB. Specifically, melt is the highest in DJF, but snowfall is typically the lowest in DJF (Lenaerts et al., 2012). Please consider evaluation of winter (or other seasons of SMB relevance)

or add language justifying the importance of summer SMB.

First of all, we would like to remind that annual SMB and surface melt rates (not only summer) are evaluated with respect to observations (Fig.4 and Fig.6). Then, we focus on a single season (summer) to analyze the teleconnections and associated mechanisms because the modes of variability and their teleconnections to the Amundsen Sea region both have strong seasonal characteristics, so that each season needs to be considered separately. We thought that analyzing all seasons separately would make the paper way too long, while showing the similarities and differences between the melt and SMB summer teleconnections was interesting. We agree that summer SMB is weaker than in other seasons, but it still represents 15% of the annual SMB (over the Amundsen Sea drainage basins) which is not negligible (vs 31%, 28% and 25% for MAM, JJA and SON respectively). The seasonal predictability of summer SMB from climate mode such as ENSO can also be of interest for operational prediction and summer field work. We have nonetheless included a supplementary table providing the correlation between SMB, and SAM, ENSO, ASL for other seasons (Table S3). The justification for the summer focus has also been added to the manuscript (section 1, L.130-134).

C2 The relationship between melt and SMB is not investigated. The paper provides background on the importance of the role of melt on hydrofracture of ice shelves and potential rapid disintegration of an ice shelf, but it does not discuss the role of firn pore space. According to Table 2, nearly all of the surface melt refreezes within the firn column, so this mechanism should be introduced as well. The paper also notes that into the future there will be more snowfall and melt, but did not mention that the enhanced snowfall could potentially also provide more pore space for meltwater infiltration and refreezing. Please consider additional discussion of the role of SMB (or snowfall) on providing addition pore space for surface melt.

First, we apologize for a mistake : we omitted to mask nunataks in the basin averages, which slightly modified the values in table 2 (now updated). For all drainage basins the
runoff is indeed equal to zero, meaning that the firn is never saturated with melt water (which is a prerequisite to form runoff in our version of MAR). The minimum rate of surface melting + rainfall needed to saturate the annual snow layer (i.e. depleting all the air in the annual snow layer) can be estimated as snowfall*[water/snow]*[1- snow/ice], where snowfall is annual (in water equivalent), and  is the density of water snow and ice. Considering a fresh snow density of 300 kg.m-3 and ice density of 920 kg.m-3, this means that the sum of annual melt and rainfall rates would need to exceed 2.25 times the annual snowfall value (all being expressed in water equivalent) to saturate the annual snow layer. This does not occur in any of the drainage basins in any year, indicating that meltwater ponding and complex surface hydrological flows are unlikely to develop over the West Antarctic drainage basins with such amount of precipitation and surface melting. The rate of surface water production (rainfall + melting) would need to increase by nearly two orders of magnitude to saturate present-day annual snow layer and therefore to initiate hydrofracturing. This is possible for strong warming scenarios given the exponential temperature dependence described by Trusel et al. (2015), although snowfall is also expected to increase (Krinner et al. 2008; Agosta et al., 2013; Ligtenberg et al., 2013; Lenaerts et al., 2016; Palerme et al., 2017), requiring even more meltwater to reach saturation. This discussion has been added to section 4, L.634-642 and L.692-697.

C3 There is no discussion on the relatively small proportions of variance explained by the climate indices. For instance, over western WAIS 20-40% of the summer SMB variability can be explained by the ASL longitude; however, it explains <6% of the Abbot, Cosgrove, and Pine Island catchments. None of the indices is significantly correlated with SMB over those catchments. Thus, the impact of ASL longitude is only relevant from Thwaites moving westward. The paper should make this clear and also potentially investigate other drivers of change for the eastern catchments or at least add clarifying statements that the drivers in eastern WAIS are unknown and potentially postulate why. Along similar lines, while ASL central pressure is a clear control on all catchments, the explained variance range from 12-21%, suggesting that there are additional factors at

play when it comes to surface melt. Would investigation of multiple regression with the different indices help clarify how they interplay (for example, perhaps the combination of some movement and strengthening or weakening of the ASL is more strongly related). Please consider adding more multivariate relationships and discuss other potential influences on meltwater production since only a small portion is explained.

First of all, we have replaced NINO34 with (-SOI) throughout our paper because, as indicated by Holland et al. (2019), SOI gives slightly stronger correlations than NINO34. Following the Reviewer's suggestion, we have investigated multi-linear regression (using a least shrinkage and selection operator (LASSO, Tibshirani 1996)) of summer SMB and melt rates onto the non-dimensionalized climate indices (divided by their standard deviation). The variance explained by individual regression coefficients are very close to the ones obtained by simple correlation, but considering the entire regression clearly shows that we are able to explain a larger portion of the SMB and melt rate variance by including several indices (16-49% for SMB and 21-30% for melting). As anticipated by the Reviewer, this indicates an interplay between the different modes of variability. A column providing the correlation of the multi-linear regression has been added to Tables 3 and 4, with associated description in section 3.2, L.464-468. Even with SOI instead of NINO34 and considering multi-linear relationships, the part of explained variance never exceeds 50% of the summer melt and SMB variance. Possible reasons for this are (i) the modes of variability do not explain all the variance locally; for example, the leading EOF of SST in the Equatorial Pacific (representing ENSO) only accounts for 50 to 70% of the SST variance (e.g. Roundy, JCLI 2015), meaning that the tropical convection thought to influence Antarctica is not completely described by SOI or NINO34; (ii) assuming that a large part of the tropospheric circulation variability is explained by ENSO, SAM and ASL indices, there are reasons why the connection may be weaker for SMB and surface melting because of their non-linear dependence on sea ice and evaporation in coastal regions, the evolution of snow properties, etc; (iii) strong modulation of the southeast Pacific extratropical circulation by Rossby wave train is not only due to the existence of El Niño events but also depends on the exact

spatial distribution of deep convection in the tropical central Pacific and to the strength of the polar jet (Harangozo et al. 2004) (iv) a part of the variability of SMB and melting may be stochastic, i.e. not necessarily driven by variability with spatio-temporal coherence at large scales. We have added a paragraph in the text (section3.2 L.492-509) to mention these possible reasons for relatively little explained variance.

C4 The postulation of potential lags is not adequately investigated. The hypothesis regarding sea ice reduction and transport from the Ross Sea could be tested as MAR using the sea-ice concentration from ERA-Interim. Thus, please consider adding analysis of sea ice concentrations to support this postulation. Although not as clear cut, intrusion of marine CDW could be evaluated by looking at the effective wind stresses as done by Steig et al. just off the continental shelf and attempt to quantify a six-month lag between strong wind events and surface melt. Also, there is no mention of potential preconditioning of the snowpack/firn for melt. An additional important variable in control of surface melt in the summer is the amount of snow that fell the prior winter, and it should be added to the analysis presented and included in Table 4. This signal might not matter at all, but also could lead to misinterpretation of an ENSO lag. Please consider all potential snowpack preconditioning variables that might explain melt from year-to-year.

In the discussion, we indeed suggested the existence of a delayed response of summer SMB and surface melting to the previous winter's ENSO events. We are not able to show perfectly robust evidence because this would probably require running dedicated experiments using a global ocean/atmosphere model (e.g. pacemaker simulations in Holland et al. 2019). We have nonetheless expanded this part of the discussion based on the literature and on additional diagnostics that indicate that such physical lag is highly probable:

1- Seasonality of ENSO and Rossby wave trains: First of all, the connection between ENSO and the Amundsen sector are thought to occur through Rossby wave trains originating in the equatorial Pacific (e.g. Ding et al. 2011). Numerous observational and

modeling papers reported that austral winter and spring conditions were more favorable for Rossby wave trains to be formed and to propagate to high southern latitudes (Harangozo 2004, Lachlan-Cope and Connolley 2006, and references therein). Scott Yiu and Maycock (2019) have recently found that the poleward propagation of tropically sourced Rossby waves in summer is inhibited by the strong polar front jet in the South Pacific sector at that time of the year, which leads to Rossby wave reflection away from the Amundsen Sea region. Steig et al. (2011) also found that changes in wind stress over the Amundsen Sea had non-significant correlations to ENSO indices in austral summer, in contrast to the other seasons showing significant correlations.

2- Snow memory: In the initial draft, we only mentioned that there was no correlation between summer melt rates and snow temperature in the previous months (which could be hypothesized as El Niño events are known to warm West Antarctica in winter; Ding et al. 2011). We agree with Reviewer #1 that snowfall in winter and spring could also be thought to influence summer melt (e.g. because the amount of fresh snow affects albedo feedbacks). However, in all the basins, we find no significant correlations between summer melt rates and snow accumulated over the previous 3 months or 6 months. For example, here are some correlation values (R):

Thwaites:

R (DJF melt / DJF smb) = 0.31(p=0.06) //// R (DJF melt / SON smb) = -0.03(p=0.86) //// R (DJF melt / JJA+SON smb) = -0.01(p=0.94) ////

Pine Island:

R (DJF melt / DJF smb) = 0.48(p=0.02) //// R (DJF melt / SON smb) = -0.21(p=0.19) //// R (DJF melt / JJA+SON smb) = -0.11(p=0.50) ////

Dotson: R (DJF melt / DJF smb) = 0.48(p=0.01) //// R (DJF melt / SON smb) = 0.09(p=0.59) //// R (DJF melt / JJA+SON smb) = 0.11(p=0.53) ////

We conclude that the lag between ENSO and melt rates is not explained by preconditioning of the snowpack in previous seasons.

3- Ocean/sea-ice memory: If the lag is not explained by snow, then it has to be explained by the other slow media, i.e. the ocean/sea-ice system. Here also, the literature provides some indications. First of all, Clem et al. (2017) mentioned stronger lagged correlation between SON ENSO and DJF sea ice cover than synchronous correlation in DJF. This lag relationship was shown to affect DJF surface air temperatures over West Antarctica (warmer for El Niño phases). Pope et al. (2017) found that El Niño events developing in MAM created a dipole of sea ice anomalies, with decreased (increased) concentration in the Ross Sea (Amundsen and Bellingshausen Seas). Using a novel sea ice budget analysis, they showed that the decreased concentration in the Ross Sea was then advected eastward, reaching the Amundsen Sea in SON and DJF.

There is also another possible pathway for lagged ENSO/sea-ice relationship. The zonal wind stress over the Amundsen Sea continental shelf break is a good proxy for the transport of Circumpolar Deep Water (CDW) onto the continental shelf (Thoma et al. 2008; Holland et al. 2019). Steig et al. (2012) noted significant correlations between that wind stress and ENSO in JJA and SON but not in DJF. All these studies pointed out scales of a few months for the build up and advection of CDW on the continental shelf then into the ice shelf cavities where they produce basal melting. As stronger ice-shelf melt rates tend to decrease sea ice in this region due to the entrainment of warm CDW towards the surface (Jourdain et al. 2017; Merino et al. 2018), this deep ocean pathway may also explain a part of the lag between ENSO and DJF sea ice in the Amundsen Sea.

To complement these analyses, we have added a composite of DJF sea ice cover anomalies for El Niño events in JJA (6-month lag Fig.14). This composite is dominated by a significant negative anomaly, confirming that ENSO in austral winter has a significant effect on sea ice 6 months later, which could arguably explain the increase in humidity and favor high melt rates and high SMB. There are several possible reasons for such a lag, it could be related to the slow advection of winter sea ice anomalies

Interactive
comment

from the Ross Sea, or to the slow advection of ocean temperature anomalies (CDW) towards the ice shelves then towards the surface through the meltwater pump, but we leave the quantification of these aspects for future research.

Minor Comments : Line 16 – change to "Amundsen Sea glaciers" Done Line 58 – change "underlaying" to "underlying" Done Line 114 – remove the ';' at the beginning of the line Done Line 140 – change "estimates" to "estimate" Done Line 185-186 – remove the sentence "These data were collected over the Thwaites and Pine Island basins." as it is redundant with Lines 181-182. Done Section 2.3 : Are these indices derived from ERA-Interim for consistency with the MAR output? If not, please state that and justify their use. We have added this information in section 2.3 Line 273 – Are "overestimate" and "underestimate" confused? Shouldn't it be "The model tends to underestimate and overestimate highest and lowest wind speeds"? We agree with the Reviewer's comments and this has been modified. Line 299 – Remove "(Medley et al. 2013, 2014)" as it is already mentioned in the sentence. Done Line 382 - add "is" after "mechanism" Done Figure 10/11 – Please add in the legend that blue represents moisture convergence for clarity. Figure 10 and 11 have been changed, we now choose to show the Integrated Vapor Transport instead of humidity convergence following concerns from reviewer #2 and D. Bromwich. Paragraph beginning with 530 – Perhaps it is important to mention here that DJF makes up the smallest percentage of annual accumulation, so it is not surprising that the findings do not match Medley and Thomas. Done

Please also note the supplement to this comment:
https://www.the-cryosphere-discuss.net/tc-2019-109/tc-2019-109-AC1-supplement.pdf

———————————————

[Figure]

Fig. 1.

[Figure]

**Fig. 2.**

Low SMB Composite

High SMB Composite

(a) IVT

(b) IVT

(c) Cloud cover

(d) Cloud cover

**Fig. 3.**

[Figure]

Fig. 4.

[Figure]

**Supplement:**

Response to reviewers

**Interannual Variability of Summer Surface Mass Balance and Surface Melting in the Amundsen Sector, West Antarctica**

Marion Donat-Magnin[1], Nicolas C. Jourdain[1], Hubert Gallée[1], Charles Amory[3], Christoph Kittel[3], Xavier Fettweis[3], Jonathan D. Wille[1], Vincent Favier[1], Amine Drira[1], Cécile Agosta[2]
* * *
**Interactive comment by Reviewer #1**

Summary – The authors present a significant amount of work on MAR model validation and ultimate evaluation of drives of summer surface mass balance (SMB) and melt over the Amundsen Sea sector of Antarctica. They provide ample background of previous studies of the climate in West Antarctica, MAR model description as well as numerous observations used in the evaluation, full descriptions of the climate indices investigated, and, of course, their findings. The model reproduced temperature, wind speed, SMB, and melt intensity and days, and thus, the authors indicate that MAR is sufficient to evaluate drivers of SMB and melt in this sector. Specifically, they find that the longitudinal position of the Amundsen Sea Low (ASL) is the primary driver (relative to ENSO and the Southern Annular Mode, SAM) of SMB variability, whereas the variability in the ASL central pressure and to a lesser extent ENSO drive melt with increasing control moving westward. The SAM was not strongly related to either SMB or melt. The authors finally surmise that there might be a 6-month lag between ENSO and West Antarctic climate via either sea ice anomalies and their transport or a lag in Circumpolar Deep Water intrusions.

The paper is generally well written and very thorough (much appreciated!) and is organized in a logical fashion. Many of the insights are not necessarily new; however, the paper presents new model work and output and attempts to present a cohesive picture of drivers of West Antarctic climate. All of the analyses presented appeared appropriate and well thought out, and the tools used were appropriate. The work presented is very thorough yet easy to understand, making it a pleasure to read.

We thank Reviewer #1 for this constructive and motivating feedback. We agree with most of the following objections and have considered them in the revised manuscript.

Major Comments :

One important consideration that is missing is a description of why summer SMB is critical and how it relates to the annual SMB. December-January-February (DJF) clearly are the relevant months for surface melting, but I think there should be more discussion as to why the paper specific isolated summer SMB. Specifically, melt is the highest in DJF, but snowfall is typically the lowest in DJF (Lenaerts et al., 2012). Please consider evaluation of winter (or other seasons of SMB relevance) or add language justifying the importance of summer SMB.

First of all, we would like to remind that annual SMB and surface melt rates (not only summer) are evaluated with respect to observations (Fig.4 and Fig.6). Then, we focus on a single season (summer) to analyze the teleconnections and associated mechanisms because the modes of variability and their teleconnections to the Amundsen Sea region both have strong seasonal characteristics, so that each season needs to be considered separately. We thought that analyzing all seasons separately would make the paper way too long, while showing the similarities and differences between the melt and SMB summer teleconnections was interesting. We agree that summer SMB is weaker than in other seasons, but it still represents 15% of the annual SMB (over the Amundsen Sea drainage basins) which is not negligible (vs 31%, 28% and 25% for MAM, JJA and SON respectively). The seasonal predictability of summer SMB from climate mode such as ENSO can also be of interest for operational prediction and summer field work. We have nonetheless included a supplementary table providing the correlation between SMB, and SAM, ENSO, ASL for other seasons (Table S3). The justification for the summer focus has also been added to the manuscript (section 1, L.130-134).

The relationship between melt and SMB is not investigated. The paper provides background on the importance of the role of melt on hydrofracture of ice shelves and potential rapid disintegration of an ice shelf, but it does not discuss the role of firn pore space. According to Table 2, nearly all of the surface melt refreezes within the firn column, so this mechanism should be introduced as well. The paper also notes that into the future there will be more snowfall and melt, but did not mention that the enhanced snowfall could potentially also provide more pore space for meltwater infiltration and refreezing. Please consider additional discussion of the role of SMB (or snowfall) on providing addition pore space for surface melt.

First, we apologize for a mistake : we omitted to mask nunataks in the basin averages, which slightly modified the values in table 2 (now updated).

For all drainage basins the runoff is indeed equal to zero, meaning that the firn is never saturated with melt water (which is a prerequisite to form runoff in our version of MAR). The minimum rate of surface melting + rainfall needed to saturate the annual snow layer (i.e. depleting all the air in the annual snow layer) can be estimated as *snowfall\*[ρwater/ρsnow]\*[1- ρsnow/ρice]*, where snowfall is annual (in water equivalent), and ρ is the density of water snow and ice. Considering a fresh snow density of 300 kg.m$^{-3}$ and ice density of 920 kg.m$^{-3}$, this means that the sum of annual melt and rainfall rates would need to exceed 2.25 times the annual snowfall value (all being expressed in water equivalent) to saturate the annual snow layer. This does not occur in any of the drainage basins in any year, indicating that meltwater ponding and complex surface hydrological flows are unlikely to develop over the West Antarctic drainage basins with such amount of precipitation and surface melting. The rate of surface water production (rainfall + melting) would need to increase by nearly two orders of magnitude to saturate present-day annual snow layer and therefore to initiate hydrofracturing. This is possible for strong warming scenarios given the exponential temperature dependence described by Trusel et al. (2015), although snowfall is also expected to increase (Krinner et al. 2008; Agosta et al., 2013; Ligtenberg et al., 2013; Lenaerts et al., 2016; Palerme et al., 2017), requiring even more meltwater to reach saturation.

This discussion has been added to section 4, L.634-642 and L.692-697.

There is no discussion on the relatively small proportions of variance explained by the climate indices. For instance, over western WAIS 20-40% of the summer SMB variability can be explained by the ASL longitude; however, it explains <6% of the Abbot, Cosgrove, and Pine Island catchments. None of the indices is significantly correlated with SMB over those catchments. Thus, the impact of ASL longitude is only relevant from Thwaites moving westward. The paper should make this clear and also potentially investigate other drivers of change for the eastern catchments or at least add clarifying statements that the drivers in eastern WAIS are unknown and potentially postulate why. Along similar lines, while ASL central pressure is a clear control on all catchments, the explained variance range from 12-21%, suggesting that there are additional factors at play when it comes to surface melt. Would investigation of multiple regression with the different indices help clarify how they interplay (for example, perhaps the combination of some movement and strengthening or weakening of the ASL is more strongly related). Please consider adding more multivariate relationships and discuss other potential influences on meltwater production since only a small portion is explained.

First of all, we have replaced NINO34 with (-SOI) throughout our paper because, as indicated by Holland et al. (2019), SOI gives slightly stronger correlations than NINO34.

Following the Reviewer's suggestion, we have investigated multi-linear regression (using a least shrinkage and selection operator (LASSO, Tibshirani 1996)) of summer SMB and melt rates onto the non-dimensionalized climate indices (divided by their standard deviation). The variance explained by individual regression coefficients are very close to the ones obtained by simple correlation, but considering the entire regression clearly shows that we are able to explain a larger portion of the SMB and melt rate variance by including several indices (16-49% for SMB and 21-30% for melting). As anticipated by the Reviewer, this indicates an interplay between the different modes of variability. A column providing the correlation of the multi-linear regression has been added to Tables 3 and 4, with associated description in section 3.2, L.464-468.

Even with SOI instead of NINO34 and considering multi-linear relationships, the part of explained variance never exceeds 50% of the summer melt and SMB variance. Possible reasons for this are (i) the modes of variability do not explain all the variance locally; for example, the leading EOF of SST in the Equatorial Pacific (representing ENSO) only accounts for 50 to 70% of the SST variance (e.g. Roundy, JCLI 2015), meaning that the tropical convection thought to influence Antarctica is not completely described by SOI or NINO34; (ii) assuming that a large part of the tropospheric circulation variability is explained by ENSO, SAM and ASL indices, there are reasons why the connection may be weaker for SMB and surface melting because of their non-linear dependence on sea ice and evaporation in coastal regions, the evolution of snow properties, etc; (iii) strong modulation of the southeast Pacific extratropical circulation by Rossby wave train is not only due to the existence of El Niño events but also depends on the exact spatial distribution of deep convection in the tropical central Pacific and to the strength of the polar jet (Harangozo et al. 2004) (iv) a part of the variability of SMB and melting may

be stochastic, i.e. not necessarily driven by variability with spatio-temporal coherence at large scales. We have added a paragraph in the text (section3.2 L.492-509) to mention these possible reasons for relatively little explained variance.

The postulation of potential lags is not adequately investigated. The hypothesis regarding sea ice reduction and transport from the Ross Sea could be tested as MAR using the sea-ice concentration from ERA-Interim. Thus, please consider adding analysis of sea ice concentrations to support this postulation. Although not as clear cut, intrusion of marine CDW could be evaluated by looking at the effective wind stresses as done by Steig et al. just off the continental shelf and attempt to quantify a six-month lag between strong wind events and surface melt. Also, there is no mention of potential preconditioning of the snowpack/firn for melt. An additional important variable in control of surface melt in the summer is the amount of snow that fell the prior winter, and it should be added to the analysis presented and included in Table 4. This signal might not matter at all, but also could lead to misinterpretation of an ENSO lag. Please consider all potential snowpack preconditioning variables that might explain melt from year-to-year.

In the discussion, we indeed suggested the existence of a delayed response of summer SMB and surface melting to the previous winter's ENSO events. We are not able to show perfectly robust evidence because this would probably require running dedicated experiments using a global ocean/atmosphere model (e.g. pacemaker simulations in Holland et al. 2019). We have nonetheless expanded this part of the discussion based on the literature and on additional diagnostics that indicate that such physical lag is highly probable:

1- Seasonality of ENSO and Rossby wave trains: First of all, the connection between ENSO and the Amundsen sector are thought to occur through Rossby wave trains originating in the equatorial Pacific (e.g. Ding et al. 2011). Numerous observational and modeling papers reported that austral winter and spring conditions were more favorable for Rossby wave trains to be formed and to propagate to high southern latitudes (Harangozo 2004, Lachlan-Cope and Connolley 2006, and references therein). Scott Yiu and Maycock (2019) have recently found that the poleward propagation of tropically sourced Rossby waves in summer is inhibited by the strong polar front jet in the South Pacific sector at that time of the year, which leads to Rossby wave reflection away from the Amundsen Sea region. Steig et al. (2011) also found that changes in wind stress over the Amundsen Sea had non-significant correlations to ENSO indices in austral summer, in contrast to the other seasons showing significant correlations.

2- Snow memory: In the initial draft, we only mentioned that there was no correlation between summer melt rates and snow temperature in the previous months (which could be hypothesized as El Niño events are known to warm West Antarctica in winter; Ding et al. 2011). We agree with Reviewer #1 that snowfall in winter and spring could also be thought to influence summer melt (e.g. because the amount of fresh snow affects albedo feedbacks). However, in all the

basins, we find no significant correlations between summer melt rates and snow accumulated over the previous 3 months or 6 months. For example, here are some correlation values (R):

Thwaites:
R (DJF melt / DJF smb) = 0.31       (p=0.06)
R (DJF melt / SON smb) = -0.03     (p=0.86)
R (DJF melt / JJA+SON smb) = -0.01   (p=0.94)

Pine Island:
R (DJF melt / DJF smb) = 0.48       (p=0.02)
R (DJF melt / SON smb) = -0.21     (p=0.19)
R (DJF melt / JJA+SON smb) = -0.11     (p=0.50)

Dotson:
R (DJF melt / DJF smb) = 0.48       (p=0.01)
R (DJF melt / SON smb) = 0.09     (p=0.59)
R (DJF melt / JJA+SON smb) = 0.11 (p=0.53)

We conclude that the lag between ENSO and melt rates is not explained by preconditioning of the snowpack in previous seasons.

3- Ocean/sea-ice memory: If the lag is not explained by snow, then it has to be explained by the other slow media, i.e. the ocean/sea-ice system. Here also, the literature provides some indications. First of all, Clem et al. (2017) mentioned stronger lagged correlation between SON ENSO and DJF sea ice cover than synchronous correlation in DJF. This lag relationship was shown to affect DJF surface air temperatures over West Antarctica (warmer for El Niño phases). Pope et al. (2017) found that El Niño events developing in MAM created a dipole of sea ice anomalies, with decreased (increased) concentration in the Ross Sea (Amundsen and Bellingshausen Seas). Using a novel sea ice budget analysis, they showed that the decreased concentration in the Ross Sea was then advected eastward, reaching the Amundsen Sea in SON and DJF.

There is also another possible pathway for lagged ENSO/sea-ice relationship. The zonal wind stress over the Amundsen Sea continental shelf break is a good proxy for the transport of Circumpolar Deep Water (CDW) onto the continental shelf (Thoma et al. 2008; Holland et al. 2019). Steig et al. (2012) noted significant correlations between that wind stress and ENSO in JJA and SON but not in DJF. All these studies pointed out scales of a few months for the build up and advection of CDW on the continental shelf then into the ice shelf cavities where they produce basal melting. As stronger ice-shelf melt rates tend to decrease sea ice in this region due to the entrainment of warm CDW towards the surface (Jourdain et al. 2017; Merino et al. 2018), this deep ocean pathway may also explain a part of the lag between ENSO and DJF sea ice in the Amundsen Sea.

To complement these analyses, we have added a composite of DJF sea ice cover anomalies for El Niño events in JJA (6-month lag Fig.14). This composite is dominated by a significant negative anomaly, confirming that ENSO in austral winter has a significant effect on sea ice 6 months later, which could arguably explain the increase in humidity and favor high melt rates and high SMB. There are several possible reasons for such a lag, it could be related to the slow advection of winter sea ice anomalies from the Ross Sea, or to the slow advection of ocean temperature anomalies (CDW) towards the ice shelves then towards the surface through the meltwater pump, but we leave the quantification of these aspects for future research.

Minor Comments :

Line 16 – change to "Amundsen Sea glaciers"

Done

Line 58 – change "underlaying" to "underlying"

Done

Line 114 – remove the ';' at the beginning of the line

Done

Line 140 – change "estimates" to "estimate"

Done

Line 185-186 – remove the sentence "These data were collected over the Thwaites and Pine Island basins." as it is redundant with Lines 181-182.

Done

Section 2.3 : Are these indices derived from ERA-Interim for consistency with the MAR output? If not, please state that and justify their use.

We have added this information in section 2.3

Line 273 – Are "overestimate" and "underestimate" confused? Shouldn't it be "The model tends to underestimate and overestimate highest and lowest wind speeds"?

We agree with the Reviewer's comments and this has been modified.

Line 299 – Remove "(Medley et al. 2013, 2014)" as it is already mentioned in the sentence.

Done

Line 382 - add "is" after "mechanism"

Done

Figure 10/11 – Please add in the legend that blue represents moisture convergence for clarity.

Figure 10 and 11 have been changed, we now choose to show the Integrated Vapor Transport instead of humidity convergence following concerns from reviewer #2 and D. Bromwich.

Paragraph beginning with 530 – Perhaps it is important to mention here that DJF makes up the smallest percentage of annual accumulation, so it is not surprising that the findings do not match Medley and Thomas.

Done
* * *
**Interactive comment by Anonymous Referee #2**

This paper presents results from the regional climate model MAR run for the Amundsen Sector of West Antarctica. The paper is well written and thorough, although some parts the paper needs improvement.

After reading the paper and collecting my points of concerns, I've read the other review and the comment of David Bromwich. I agree with their major concerns and these concerns have to be addressed. Additionally, I have the following major comments:

It needs to be addressed why SMB summer is discussed and not the annual SMB. I can imagine a reason, but this - or any other - reason is not given.

We thank Reviewer #2 for these constructive and motivating feedbacks. As for Reviewer#1 we agree with most of the objections and have considered them in the revised manuscript. As detailed in our response to Reviewer #1, we have added an explanation on why we focused on summer SMB (now in section 1).

Although the patterns in Figures 8-12 are logic and reasonable, its worrisome that most of the signals showed are insignificant. Try to get a better understanding of the significance. For example, for geopotential fields the gradient matters more than the value, so you might take a "relative elevation" approach similar to the ASL central pressure. You might also try a different method to determine significance, for example, bootstrapping. If the patterns remain mostly insignificant it implies that the shown patterns do occur during high/low melt/SMB but not necessarily lead to high/low melt/SMB.

We first would like to point out that all the composites showed a clear significant area over the coastal region of the Amundsen Sea and over the studied drainage basins, to the exception of the 700 hPa geopotential height and the humidity divergence at 850 hPa. We thank the Reviewer for his suggestion that the mean geopotential height matters less than the gradient.

To circumvent this issue, we have added the composite by analyzing the geopotential pattern divided by the domain-averaged value for each DJF season. This produces much more significance than in the initial version, as shown in the modified Fig. 9 and Fig. 12. See next comment for the case of humidity divergence.

I'm not convinced that humidity convergence @ 850 hPa is the best parameter to show. For SMB anomalies: As the moisture holding capacity of air is not that big, the convergence is directly linked to precipitation generation. Added compared to SMB is a whole bunch of noise due to variations in the elevation of the 850 hPa level and noise is added by apparently near stationary numerical waves. I would be more interested to see anomalies in the temperature @ 700 hPa / 850 hPa and vertical integrated moisture content fields. For melt anomalies: it likely boils down to that high melt years have also higher summer SMB although this relation might not be significant. Furthermore, the authors do show that cloudiness increases, but fail to prove that his is the only cause. To which extend is the higher melt due to cloudiness and which extend due to advection of warmer air? What is the anomaly of temperatures at 700 hPa? This anomaly can be easily included in Figures 12 a,b. I know temperature and cloudiness anomalies are likely covarying, so disentangling might be complicated. Helpful might be the MSSA technique (Plaut and Vautard, 1994; Allen and Robertson, 1996).

We agree that the divergence of humidity transport was too noisy and, in the end, little supportive of our mechanism. We have replaced this diagnostic by the integrated vapor transport (IVT) that is calculated as:

$$IVT = \int_{925}^{700} qv \frac{dP}{g}$$

where $v$ is the velocity along the y-axis and $g$ the gravity parameter. We have replaced humidity divergence composites by IVT composites in Fig.10 and Fig.11. It clearly shows that high SMB and high melt rates are linked to a strong southward vapor transport towards the drainage basins of the Amundsen Sea. The arrival of this vapor from the mid-latitude into the colder Antarctic region can arguably induce condensation and cloud formation.

We have also looked at the composite of sensible heat fluxes (this has been added to the supplementary material) versus longwave downward heat fluxes. The sensible heat flux is negative for the high-melt composite, which means that energy is going from the snow surface to the air (the temperature of the snow surface is higher than the temperature of the air above), so advection of warm air above the surface is not responsible of higher surface melt. Therefore, as suggested in our initial manuscript, changing downward longwave heat flux is the main mechanism for low/high surface melt events. Increase in cloudiness and humidity transport is therefore the main driver. We have added a comment on this in section 3.2, L. 418-419. Fig.10 and Fig.11 has been changed. (Figure below has been added to supplementary material)

(line 527): CDW intrusions cannot be proven directly with the data from this manuscript (although SSTs and wind stress are available), but sea ice anomalies are available. It takes only a few steps to verify if the hypotheses are confirmed by data, so take those steps. And if the data does not confirm this hypothesis, that must be stated as well.

See our response to Reviewer #1: we have substantially expanded our discussion of this hypothesis based on further literature review and on an additional DJF sea ice composite for JJA Niño events. Although providing perfectly robust evidence of causality would require specific AOGCM experiments, we believe that several lines of evidence indicate that such physical lag is highly probable.

Minor comments :

158: The sentence on the boundary relaxation is ambiguous: It could also mean that every 6 hours the state in at the boundaries "is forced back" to ERA-Interim values. However, I presume that every time step fields are relaxed to ERA-Interim fields with 6- hourly temporal resolution. Rephrase to remove this ambiguity. Furthermore, add the boundary relaxation zone to the graph by using shading or something else and explain in the text how wide this zone was. From eg Fig 10 I conclude it was rather narrow, explain why or add a reference.

Explanation about boundary relaxation has been changed, as well as Fig.1 where relaxation zone is now shown in white. Fig.10 has been change related to next comments.

131: polar-oriented. Did you mean "polar adapted"? Oriented is not wrong but uncom- mon in this meaning.

Done

224: I would prefer if these webpage-links could be included as references so that the text becomes less disturbed. But that's up to Copernicus to solve/decide on.

We followed the manuscript preparation guidelines for authors (webpage, references)

298: How this performance compares to other studies, thus MAR-full Antarctica and various RACMO2 products? Add a comment on this in the text.

We have added a comment in section 3.1 L-287.288. Compared to MAR-full Antarctica we present very similar biases (Agosta et al., 2019), correlation for SMB compared to observation (Glacioclim SAMBA). R=0.95 for our simulation and R=0.93 for MAR-full Antarctica, same for bias of 0.13 and 0.14 for MAR-full Antarctica. This improvement is not really significant and can be explained only by higher resolution and higher spatial variability in our simulation. Comparison with RACMO2 products is beyond the scope of our studies and will be the subject of future studies.

321: It might be interesting to make a scatter plot of the modelled and interpolated QuickScat melt for their overlapping time period. You could color code the dots per ice shelve or drainage

basin and even add regression lines per drainage basin. You don't discuss the few spots in West Antarctica where MAR gives high melt rates – do this. And have a look at https://www.the-cryosphere.net/13/1473/2019/tc-13-1473-2019.pdf if this might be a possible explanation for your model deviations too.

We agree we have added scatter plot (Fig.6) and related discussion L.336-337.

359: It would be nice if these high/low SMB/melt years as listed, maybe by adding symbols in figure 7.

We have added a table in supplementary material (Tab.S4) and not in Fig.7 because composite dates correspond to values with low/high SMB/Melt only over Thwaites and Pine Island basins and not over the model domain as explained in section 3.2 "For a sake of clarity, we only consider the Pine Island and Thwaites basin (together) as a first approach. "

363: In Figure 8 your plotting two differences per frame – that makes it harder to include signs of significance. Are these differences significant? Make a comment in the text and, if possible, find a way to display if deemed relevant.

Fig.8 does not represents differences (composite - climatology) like other composite as the climatology is shown in grey and composite in color, we choose to plot all in the same panel as the comparison of both the high and low composites with the climatology are necessary.

378: Cloud cover could be a poorly performing parameter – I know models in which this is the case. Verify if you find similar/equivalent patterns in the vertical integrated cloud content (please add these figures in the rebuttal letter) and state in the manuscript if similar / equal patterns are found in the vertical integrated cloud content.

We have added IVT (integrated vapor transport) in Fig.10 and Fig.11 and here is the composite for integrated water vapor (kg m$^{-2}$):

[Figure]

379: As snowfall exceeds the SMB due to sublimation, the "95%" in the quote is a bit odd. Rephrase.

Done

425-427: This is not necessarily true. If positive SMB anomalies occur only if NINO34 is positive and ASL-longitude is negative, then their impact on SMB is not unrelated even though NINO34 and ASL-longitude are unrelated themselves

We do not understand the reviewer's concern with our sentence ("NINO34 and the ASL longitudinal location are not significantly connected together (Table 1), therefore their connection to SMB can be considered as independent from each other"). In the example given by the reviewer, both ENSO and the ASL migration impact SMB, but both act through independent connections (assuming linear relationships, i.e. perfectly described by correlations). It does not mean that the coincidental phases of the two indices do not explain the strongest SMB events.

438: Would it not be more straightforward to see if there is a correlation between SMB and melt rates? And if not, state this.

See our response to the major comments by Reviewer#1

Table S1: Add the numbers used in Fig 1 to the table – Yes, I know they are ordered from 1 to 41, but adding the number makes it just a slightly bit easier

We agree, this has been done

Fig 1: Consider to include excluded AWS stations in the figure using a different color, as long as they are on the map. Names are not needed.

We think that display the 243 AWS can disturb the understanding of the figure, so we have not followed this suggestion.

Fig 2: The lines are not explained in the figure caption. Are the lines derived using normal fitting or perpendicular fitting techniques? Colors are not different enough to identify stations in the graph, so either use more distinguishable colors or simply don't try: give all lines the same color.

We use least-mean-square fit (linregress in python) we have added this information within the caption.

A drawback of a dot-plot is that you can't see differences in density once the dots form a continuous cloud. It might be worth the work to calculate the dot-density per (e.g.) 0.01 C-squared (Fig 2a) and plot this point density as contour graph on top of the dots. This added information on the point-density would make a statement like line 273-274 visible from the graph, the overestimation for low wind speeds is not well visible in the point cloud.

We have changed Fig.2 and have added transparency on dots in order to see density differences. We have kept stations names (some might it useful) but we have changed colors (less transparent, more distinguishable). Legend has been changed and fitting method explained.

Fig 3: I'm not fond of the graphical solution to plot SMB in greyscale – details are hardly visible nor quantifiable. For example, I have no clue what the magnitude of the SMB from MAR is near the Medley data. Replace the grey by colors and add the basin delineation in a different manner. In all solutions, more detail must become visible for SMB ranging from 200 to 500 mm w.e. per year.

We agree and we have changed Fig.3 (colormap, and range).

Fig 5: Replace the grey by clear colors and extend the scale to higher values than 100 mm w.e. per year – this should be obvious as you do discuss these high melt values in the main text.

We agree with this comment and we have changed the grey colors. As far as the color bar is concerned, we discuss high melt values but only over Thwaites and Pine Island, that's why we choose a color bar where differences in melt rate over Thwaites and Pine Island are distinguishable.

Fig 9: Contours in b are labelled with 0-2-5-8 intervals, but their regular spacing looks like 0-2.5-5-7.5. Check this. Hatching is not explained – should be done here too. Hatching line thickness varies with viewer.

Fig.9 has been changed and contours checked. We have added hatching explanation.

Plaut, G., and R. Vautard, 1994: Spells of low-frequency oscillations and weather regimes in the Northern Hemisphere. J. Atmos. Sci., 51, 210–236, doi:https://doi.org/10.1175

Allen, M. R., and A. W. Robertson, 1996: Distinguishing modulated oscillations from coloured noise in multivariate datasets. Climate Dyn., 12, 775–784, doi:https://doi.org/10.1007
* * *
**Interactive comment by David Bromwich**

This is a comprehensive analysis.

We thank David Bromwich for these comments that have pushed us to improve our manuscript.

One major shortcoming is that it really underplays the comparisons of the present results with those obtained by Deb et al. (2018) also based on regional climate modeling for 1979-2015 summers where a leading conclusion is: "El Niño episodes during austral summer drive warmer conditions over Amundsen Sea Embayment ice shelves that cause enhanced surface melting". El Niño influences play a relatively minor role in the current analysis. The explanation likely lies in the discussion on lines 475-486.

Deb et al. (2018) was cited 4 times in the submitted manuscript, but going back to our text, we agree that there should be more comparison to the results obtained by Deb et al. (2018) about the connections to ENSO and the ASL. We have added that "The relationship between ENSO and the number of melt days was identified by Deb et al. (2018) using both regional simulations and a satellite product". Our results are difficult to compare more quantitatively because different methods and metrics were used in Deb et al. (2018) and in our study. We now also mention that "longitudinal migrations of the ASL are not the main driver of surface melting variability, as previously noted by Deb et al. (2018)".

I didn't think the analysis for a lagged relation between El Niño forcing SMB/melting (Fig. 13) to be very compelling, at best possible.

See our response to Reviewer #1: we have substantially expanded our discussion of this hypothesis based on further literature review and on an additional DJF sea ice composite for JJA Niño events. Although providing perfectly robust evidence of causality would require specific AOGCM experiments, we believe that several lines of evidence indicate that such physical lag is highly probable.

I don't understand what is meant by humidity divergence (Figs. 10 and 11). Normally one evaluates moisture transport divergence in relation to P-E. Please clarify.

We apologize for the lack of clarity and we did plot the moisture transport divergence. We have nonetheless decided to show the meridional integrated vapor transport instead of the divergence, which better shows the moisture transport from the mid-latitudes to the Antarctic ice sheet.

References that were not cited in the initial draft:

Clem, K. R., Renwick, J. A., and McGregor, J. (2017). Large-scale forcing of the Amundsen Sea Low and its influence on sea ice and West Antarctic temperature. Journal of Climate, 30(20), 8405-8424.

Ding, H., Greatbatch, R. J., and Gollan, G. (2014). Tropical influence independent of ENSO on the austral summer Southern Annular Mode. Geophysical Research Letters, 41(10), 3643-3648.

Harangozo, S. A. (2004). The relationship of Pacific deep tropical convection to the winter and springtime extratropical atmospheric circulation of the South Pacific in El Niño events. Geophysical research letters, 31(5).

Holland, P. R., Bracegirdle, T. J., Dutrieux, P., Jenkins, A., and Steig, E. J. (2019). Climate Forcing of the West Antarctic Ice Sheet: Anthropogenic Trends and Internal Climate Variability. Nature Geoscience.

Lachlan-Cope, T. and Connolley, W. (2006). Teleconnections between the tropical Pacific and the Amundsen-Bellinghausens Sea: Role of the El Niño/Southern Oscillation. Journal of Geophysical Research: Atmospheres, 111(D23).

Roundy, P. E.: On the Interpretation of EOF Analysis of ENSO, Atmospheric Kelvin Waves, and the MJO, J. Climate, 28(3), 1148–1165, doi:10.1175/JCLI-D-14-00398.1, 2014.

Scott Yiu, Y. Y. and Maycock, A. C. (2019). On the seasonality of the El Niño teleconnection to the Amundsen Sea region. Journal of Climate.

Thoma, M., Jenkins, A., Holland, D. and Jacobs, S. (2008). Modelling circumpolar deep water intrusions on the Amundsen Sea continental shelf, Antarctica. Geophysical Research Letters, 35(18).

---

## Author Comment (AC2) · 27 Sep 2019

We thank David Bromwich for these comments that have pushed us to improve our manuscript.

C1 :One major shortcoming is that it really underplays the comparisons of the present results with those obtained by Deb et al. (2018) also based on regional climate modeling for 1979-2015 summers where a leading conclusion is: "El Niño episodes during austral summer drive warmer conditions over Amundsen Sea Embayment ice shelves that cause enhanced surface melting". El Niño influences play a relatively minor role in the current analysis. The explanation likely lies in the discussion on lines 475-486.

[Figure]

Deb et al. (2018) was cited 4 times in the submitted manuscript, but going back to our text, we agree that there should be more comparison to the results obtained by Deb et al. (2018) about the connections to ENSO and the ASL. We have added that "The relationship between ENSO and the number of melt days was identified by Deb et al. (2018) using both regional simulations and a satellite product". Our results are difficult to compare more quantitatively because different methods and metrics were used in Deb et al. (2018) and in our study. We now also mention that "longitudinal migrations of the ASL are not the main driver of surface melting variability, as previously noted by Deb et al. (2018)".

C2: I didn't think the analysis for a lagged relation between El Niño forcing SMB/melting (Fig. 13) to be very compelling, at best possible.

See our response to Reviewer #1: we have substantially expanded our discussion of this hypothesis based on further literature review and on an additional DJF sea ice composite for JJA Niño events. Although providing perfectly robust evidence of causality would require specific AOGCM experiments, we believe that several lines of evidence indicate that such physical lag is highly probable.

C3: I don't understand what is meant by humidity divergence (Figs. 10 and 11). Normally one evaluates moisture transport divergence in relation to P-E. Please clarify.

We apologize for the lack of clarity and we did plot the moisture transport divergence. We have nonetheless decided to show the meridional integrated vapor transport instead of the divergence, which better shows the moisture transport from the mid-latitudes to the Antarctic ice sheet.

Please also note the supplement to this comment:
https://www.the-cryosphere-discuss.net/tc-2019-109/tc-2019-109-AC2-supplement.pdf

---

## Author Comment (AC5) · 27 Sep 2019

The comment was uploaded in the form of a supplement:
https://www.the-cryosphere-discuss.net/tc-2019-109/tc-2019-109-AC5-supplement.pdf

---

## Author Comment (AC6) · 27 Sep 2019

The comment was uploaded in the form of a supplement:
https://www.the-cryosphere-discuss.net/tc-2019-109/tc-2019-109-AC6-supplement.pdf

---

## Author Comment (AC7) · 27 Sep 2019

The comment was uploaded in the form of a supplement:
https://www.the-cryosphere-discuss.net/tc-2019-109/tc-2019-109-AC7-supplement.pdf

---

## Author Comment (AC8) · 27 Sep 2019

The comment was uploaded in the form of a supplement:
https://www.the-cryosphere-discuss.net/tc-2019-109/tc-2019-109-AC8-
supplement.pdf

---

## Author Comment (AC9) · 27 Sep 2019

The comment was uploaded in the form of a supplement:
https://www.the-cryosphere-discuss.net/tc-2019-109/tc-2019-109-AC9-supplement.pdf

---

## Author Response (AR2)

**Interannual Variability of Summer Surface Mass Balance and Surface Melting in the Amundsen Sector, West Antarctica**

Marion Donat-Magnin[1], Nicolas C. Jourdain[1], Hubert Gallée[1], Charles Amory[3], Christoph Kittel[3], Xavier Fettweis[3], Jonathan D. Wille[1], Vincent Favier[1], Amine Drira[1], Cécile Agosta[2]
* * *
[Comments from the editor]

I have an overall impression that the manuscript needs careful proof reading. For example, SMB and DJF need to be used more consistently, rather than both SMB and surface mass balance, and both DJF and December-January-February are used in an interchangeable manner. Also, some figure captions are incomplete. All line numbers below show these in the markup manuscript.
We thank the editor for taking the time to evaluate our revised manuscript, and we apologize for these remaining mistakes.

L139: Define DJF here as it appears first time here. It is currently defined at L248.
Done (L.131 of revised manuscript).

L203: MAR has 10 km horizontal resolution (L169) so all AWS in your model domain can be located less than 15 km from the MAR grid points. So, I think that first criterion is always met. Please clarify. And show how many AWS are located in the model domain (out of 243 over Antarctica).
We have added more explanation about this: "even if the domain resolution is 10 km, stations over islands or capes that are not resolved can be located farther than 15 km from the closest continental MAR grid point" (L.181-182). We have also added the numbers of stations used (41 out of the 243 available over Antarctica).

L213-214: do you want to say that reflectors detected with airborne radar are dated using firn cores?
We want to say that SMB derived from airborne radar was validated against firn cores. This has been clarified L. 192-193.

L215: Show time span of SMB values in the GLACICLIM SAMBA dataset.
We completed the sentence L.194-201.

L251-254: revise to "(as also found by Scott al., 2019 and Holland et al. 2019). Our analysis found very similar results using NINO3.4 (not shown)." Currently it is unclear who obtained the results with NINO3.4. And "3.4" or "34"?
This has been corrected following the editor's suggestion.

L328: define RMSE here (its definition in the figure caption is inadequate).
We now define RMSE L.284. The 10th and the 90th percentile are related to all RMSE. (i.e. the 10th percentile and 90th of the RMSE series for all the 41 stations)

L334-335: Why are less satisfactory results in the inland explained by more flatter, smoother topography there?
There seems to be a misunderstanding: we are speaking about island and resolution : 10 km is still too coarse to resolve topographic features of the smallest islands.

L337: change to DJF (also in Fig. 2 caption and elsewhere; once it is defined use it consistently).
Done.

L931: make a new paragraph here?
Done.

L992: Medley and Thomas (2019), not 2009.
Sorry for this mistake, we have changed the year to 2019.

Table 2: where is it cited? Please make sure that all tables are cited in the right order.
We have added a sentence L.325-326.

Figure 1: add distance bar.
Done.

Figure 2, Table S1, Table S2: please clarify that these are daily values (it is said daily at L326 but not explicit enough in the figure and tables).
Done.

Figure 3: update the caption. Drainage basins are not shown in the revised figure.
The drainage basins are shown using large grey contours in the revised version, as described in the caption "The drainage basins under consideration are the same as in Fig.1 (large grey contours here)."

Figs 9, 10, 11: panel a shows IVT along the y axis of this map projection. Then you need to define the map projection used in this paper.
We add the projection (oblique stereographic projection EPSG:3031) L.156

Figure 9: add scale of arrow lengths or at least explain it in the caption. Also explain hatched area more clearly. What kind of significance? And which panels show these hatches (all a-f or only a-d)?
We have added in the caption that scale of arrow lengths are shown near the upper right corners of panels (e) and (f). And we now mention that hatches are shown for panels (a-d).

Figure 10: define q, v, p, and g. Similar to Figure 9, clarify this t test and significance.
Done.

Figure 11: change "same formula as for SMB" to "same formula as for Fig. 10". Similar to
Done.

Supplement:
Table S1: typo. Change S to S1.
Done.
* * *
[Comments from reviewer#1]

The paper has undergone significant revision and the authors have made several improvements. The new version is generally well-written, including a thorough description of relevant processes, and the version is clearly the result of a substantial amount of work.

Minor Comments

Line 220: Why not directly calculate the SAM index from the MAR pressure fields (and actually, the same for the ASL?)
To calculate the SAM a global simulation is needed. Our domain is still too small to calculate the ASL (sector pressure is defined as area-average sea level pressure over sector 170-298°E : 60-80°S, Hosking et al., 2016)

Line 318: The details of the comparison with the SAMBA database should be improved. Those measurements cover lots of different time windows, how was that considered? If it wasn't, it should be mentioned that that is a possibility for the decreased performance as compared to the snow radar data.
Details are now given L.194-201.

Table 3 and 4: The multiple regression is an interesting addition. It would be more useful if the regression coefficients as well as their individual p-values for each index are listed, so the reader can understand how they interplay.
Computing p-values for the LASSO is difficult, because the optimization problem of the LASSO introduces a selection procedure on the variables, setting some to zero and some not to zero based on their correlations (see https://arxiv.org/pdf/1311.6238.pdf : "The problem with this is that the p-values can no longer be trusted, since the variables that are selected will tend to be those that are significant")

Line 690: I think care should be taken to also mention the strength of this signal (or lack thereof) as the lagged explained variance still remain very small.
We agree and have added the following sentence: "It should nonetheless be noted that even accounting for this 6-month lag, the influence of ENSO on summer SMB and melt rates remains weak, not explaining more than 15% variance" (L. 709-711).

Technical Comments

Page 1, Line 24: consider revising sentence to: Forty percent of the interannual summer SMB variance over the Getz Ice Shelf is explained by the westward ASL longitudinal migrations
We lose an important result with the reviewer suggestion: the percentage increase westward toward Getz, we therefore suggest to keep the sentence as before.

Line 66: Change "Understand" to "Understanding"
Done.

Line 155: Change to "Amundsen Sea Embayment glaciers"
Done.

Line 163: remove the extra comma and parenthesis after Dee et al.

Done.

Line 346: change "pronounce" to "pronounced"
Done.

Line 348: Please explain what variable it is the MAR overestimates. It's not clear if its melt or snowfall
Thanks for this comment, it was about surface melt, we have completed L.355.

Line 400: change to Amundsen Sea sector
Done.
* * *
[Comments from reviewer#2]

The authors faithfully addressed my concerns and apart from the minor comments below, I think this paper is ready for publication.
L177: The text state ~400 km but the map shows a white zone of about 50 km. Which of the two is true? Or does the domain extents further than Figure 1? In any case, something needs to be adjusted.
Sorry for this mistake, the true value is 50km. We adjusted the text.

Figure 5: The authors provides an argument to stop the color scale at 100 mm w.e. per year. I'm not convinced, the other ice shelves are also discussed although in lesser extend. Moreover, I simply do not see the necessity to cut off at 100 mm w.e. per year as it is doable to expand to ~200 mm w.e. per year without loss of clarity. They seem to use now something similar to ncl's color table cmocean_tempo (thus white, light green, blue, black), but if you use a colormap/color table with more colors, you can show the values up to 200 without losing "signal" for the 10 mm w.e. per year values.
We agree, we have changed figure 5 as suggested.

Figure 9c/d & 12c/d: These are a good additions, but it should be explained better what is exactly shown in these panels. The logical place for this explanation is in the running text.
We have added some explanations L.405-410 (about Fig. 9) and L. 424-428 (about Fig. 12).

Figure 13: adjust scale of the y-axis that dotted line in e doesn't get cut off.
Fig. 13 has been modified as suggested.

Line 990: would "… austral summer, which represents 15% of the annual SMB, and correlations…" not be grammatically better?
Thanks for this comment, L.990 (L.629 in the revised manuscript) has been changed.

Line 1029: "We now discuss" is a bit odd, consider rephrasing into something like "Lastly, we discuss".
Done.

Line 1030 The assumption that you need melt+rainfall to be 2.24 times snowfall before the snowpack saturates conflicts with earlier estimates. Pfeffer, 1991, JGR estimated it to be 0.7, recent observations align with that. I know, that is not saturation with water, but simply filling the pore space with refrozen water. Furthermore, Kuipers Munneke, JoG, doi: 10.3189/2014JoG13J183, made an estimate of when this process may cause ice-shelf collapse. Ice shelf collapse in this region is not as unlikely as the authors suggest now.

We thank the reviewer for pointing to Pfeffer et al. (1991), and there was indeed a mistake in our estimation: our ratio of 2.24 is valid for the case of rainfall bringing extra water into the firn, but in the case of melting, the liquid water is not extra water, it is removed from the snow mass, and the ratio becomes 0.69. Pfeffer et al. assume a maximum saturation of 830 kg/m$^3$ (due to some kind of close off), which reduces the ratio to 0.64, but they also take into account a melt quantity to warm the firn to 0°C (required to store liquid water), which in the end gives 0.70. To keep it simple, we have modified our text and now refer to an approximate ratio of 0.7 according to Pfeffer et al. (1991). The calculation is now done only over ice-shelve (instead of all drainage basin as before) which is more pertinent for hydrofracturing. Snowfall and surface melt over individual ice shelves has been added to Table.2.

We have added a reference to Kuipers Munneke et al. at the end of the conclusion: "In their projections, Munneke et al. (2014) found that the Western part of Abbot as well as Cosgrove could become water-saturated before the end of the 22$^{nd}$ century, but the other ice shelves of the Amundsen sector remained non-saturated".

Line 1090: rephrase, see comment at 1030.
Done.